# Expansion of human megakaryocyte-biased hematopoietic stem cells by biomimetic Microniche

Yinghui Li [1,2,6], Mei He [1,2,6], Wenshan Zhang[1,2,6], Wei Liu[3,4,6], Hui Xu[1,2,6], Ming Yang[1,2], Hexiao Zhang[1,2], Haiwei Liang[3], Wenjing Li[3], Zhaozhao Wu[3], Weichao Fu[1,2], Shiqi Xu[1,2], Xiaolei Liu[1,2], Sibin Fan[1,2], Liwei Zhou[1,2], Chaoqun Wang[1,2], Lele Zhang[1,2], Yafang Li[1,2], Jiali Gu[1,2], Jingjing Yin[1,2], Yiran Zhang[1,2], Yonghui Xia[1,2], Xuemei Mao[5], Tao Cheng [1,2] ✉, Jun Shi[1,2] ✉, Yanan Du [3,4] ✉ & Yingdai Gao [1,2] ✉

Limited numbers of available hematopoietic stem cells (HSCs) limit the widespread use of HSC-based therapies. Expansion systems for functional heterogenous HSCs remain to be optimized. Here, we present a convenient strategy for human HSC expansion based on a biomimetic Microniche. After demonstrating the expansion of HSC from different sources, we find that our Microniche-based system expands the therapeutically attractive megakaryocyte-biased HSC. We demonstrate scalable HSC expansion by applying this strategy in a stirred bioreactor. Moreover, we identify that the functional human megakaryocyte-biased HSCs are enriched in the $CD34^+CD38^-$ $CD45RA^-CD90^+CD49f^{low}CD62L^-CD133^+$ subpopulation. Specifically, the expansion of megakaryocyte-biased HSCs is supported by a biomimetic niche-like microenvironment, which generates a suitable cytokine milieu and supplies the appropriate physical scaffolding. Thus, beyond clarifying the existence and immuno-phenotype of human megakaryocyte-biased HSC, our study demonstrates a flexible human HSC expansion strategy that could help realize the strong clinical promise of HSC-based therapies.

Hematopoietic stem cells (HSCs) are understood as fundamental for realizing the well-recognized, vast potential of HSC-based therapies in medicine, and for research and treatment in diverse areas such as immune system disorders, both malignant and non-malignant hematological disorders, hemoglobinopathies, and myeloproliferative disorders[1–4]. Functional HSCs with self-renewing and multi-lineage reconstruction capabilities are essential for successful HSC transplantation[5]. On one hand, the need for large quantities of donor HSCs have been important driving forces behind the study of HSC expansion. On the other hand, the problem of lineage recovery failure in HSC-based therapy, especially the Mk lineage, have called the demand for the study of HSCs function maintenance[6]. Patients who fail

[1]State Key Laboratory of Experimental Hematology, National Clinical Research Center for Blood Diseases, Haihe Laboratory of Cell Ecosystem, PUMC Department of Stem Cell and Regenerative Medicine, CAMS Key Laboratory of Gene Therapy for Blood Diseases, Institute of Hematology and Blood Diseases Hospital, Chinese Academy of Medical Sciences & Peking Union Medical College, Tianjin 300020, China. [2]Tianjin Institutes of Health Science, Tianjin 301600, China. [3]Department of Biomedical Engineering, School of Medicine, Tsinghua-PKU Center for Life Sciences, Tsinghua University, 100084 Beijing, China. [4]Beijing CytoNiche Biotechnology Co. Ltd., 100195 Beijing, China. [5]Nankai Hospital, Tianjin Hospital of Integrated Traditional Chinese and Western Medicine, Tianjin 300100, China. [6]These authors contributed equally: Yinghui Li, Mei He, Wenshan Zhang, Wei Liu, Hui Xu. ✉e-mail: chengtao@ihcams.ac.cn; shijun@ihcams.ac.cn; duyanan@tsinghua.edu.cn; ydgao@ihcams.ac.cn

in Mk lineage reconstitution after transplantation need to rely on platelet transfusion, which is a heavy financial and physiological burden[7–9]. However, the strategy of "high-fidelity" expansion of HSCs with self-renewal and Mk-lineage reconstruction ability has yet to be optimized.

Recent studies have revealed significant HSC heterogeneity, including evidence for early HSC lineage segregation in addition to the presence of lineage-biased HSCs and lineage-restricted progenitors within the HSC compartment[10–13]. The multipotent but megakaryocyte (Mk)/platelet-biased HSCs are considered directly function to Mk lineage reconstruction, which is difficult but necessary in HSC-based therapies[14]. Studies revealed that Mk-biased HSCs resided at the apex of the HSCs hierarchy, and identified the markers of Mk-biased HSCs based on simultaneous localization and mapping (SLAM) long-term HSC makers in mice[11,13]. It has been reported that megakaryocytes are directly derived from hematopoietic stem and progenitor cells with few intervening intermediates megakaryocyte-erythroid progenitors (MEP) in adult human bone marrow haematopoiesis[15], indicating the existence of human Mk-biased HSCs whose immune-phenotype remains unclear. Elucidating the immune-phenotype of human Mk-biased HSCs will contribute to the improvement of Mk lineage reconstruction in HSC transplantation.

The capacity for stable expansion of functional HSCs is essential to exploiting the full potential of HSC-based therapies. Currently, well-established culture methods incorporate small molecules (e.g., SR1, nicotinamide and UM171)[16–19], peptides (NOV)[20], and macromolecules (PVA)[21]. The expansion of human HSCs in 3D culture using nanofiber materials and zwitterionic hydrogel (ZTG) was also recently demonstrated[22,23]. However, several unresolved issues still limit the use of current techniques in a clinical setting for HSC expansion: first, the requisite sorting of CD34+ or CD133+ cells for starting material; second, Mk lineage reconstruction had not previously been evaluated due to lack of reliable in vivo evaluation models of Mk reconstruction; and third, although several studies reported scalable culture of MSCs[24], the scalable expansion of functional human HSCs was rarely reported. Thus, a reliable strategy for the expansion of human Mk-biased HSCs is still urgently needed to advance both basic research and clinical hematology applications requiring functional megakaryocytes.

In this study, we developed a biomimetic Microniche[25–28] for HSC cultures that we hypothesized could support the efficient expansion of HSCs (Fig. 1a), based on the fact that HSCs reside in the bone marrow cavity in vivo, where the natural microenvironment supports the maintenance and expansion of HSCs[29–32]. In designing this biomimetic strategy, we also considered the use of more accessible starting materials (e.g., mononuclear cells (MNCs) derived from umbilical cord blood (UCB), peripheral blood (PB), and bone marrow (BM)) for generating HSCs that are highly phenotypically similar to those produced in the natural bone marrow context. Using an in vivo Mk lineage evaluation strategy exclude the recipients' Mk cells, we demonstrated promotion of Mk reconstruction by our Microniche-based system and identified the immune-phenotype of human Mk-biased HSCs. Mechanistically, we found that both cytokine milieu and the physical scaffold generated by the Microniche contributed to the expansion of HSCs and Mk-biased HSCs. Our study demonstrates a flexible human HSC expansion strategy that should help achieve the strong clinical potential of diverse HSC-based therapies.

## Results

### A Microniche-based culture system can support expansion of human HSCs

The Microniche culture system was specifically developed to resolve the long-standing issues of lost proliferative and differentiative capability during expansion of HSCs. Microniche could be applied as a tablet that rapidly dissolves in buffer or culture medium to produce a colloidal suspension of tens of thousands of Microniche scaffolds

(Supplementary Fig. 1a–d). Scanning electron microscopy and florescent imaging confirmed the internal and external distribution of HSCs in the Microniche scaffolds (Fig. 1b). Flow cytometry was then used to evaluate the effects of Microniche on expansion of phenotype-defined hematopoietic stem and progenitor cells (HSPCs) subpopulations[33] (Supplementary Fig. 1d). Culture with conditioned medium (StemSpan SFEMII + 1% P/S + 10 ng/ml hSCF + 100 ng/ml hTPO) only was used as a control here and in subsequent experiments. Compared with the reported HSC-expanding agents SR1, UM171, and PVA, Microniche realized stronger enhancement of the CD34+CD38−CD45RA−CD90+CD49f+ cell population upon seeding UCB CD34+ cells. When MNCs from UCB, PB, or BM were seeded, Microniche group realized highest absolute number of CD34+CD38−CD45RA−CD90+ and CD34+CD38−CD45RA−CD90+CD49f+ HSC populations among these tested groups (Fig. 1c–e, Supplementary Fig. 1f and Supplementary Dataset 1). These data suggested that the Microniche-based system had a higher capacity for expansion of primitive HSCs obtained from different sources.

Limiting dilution xenograft assays in immunodeficient NOG mice (Supplementary Fig. 2a) showed that Microniche expansion resulted in a significantly greater frequency of human HSCs (5.01-fold) compared to that observed in xenografts of fresh cells, with frequencies of 1/11,994, 1/16,624, and 1/2394 for the fresh, control, and Microniche groups, respectively (Fig. 1f, Supplementary Fig. 2b and Supplementary Dataset 2). In addition, we experimentally confirmed that Microniche-derived cells could differentiate into multiple lineages and that the engrafted cells retained their capacity for HSPCs reconstruction in vivo, indicating they were unaffected by pre-graft expansion (Supplementary Fig. 2c–e). A secondary transplantation in NOG mice was performed to determine whether Microniche affected the capability for HSC self-renewal, which revealed no significant differences in engraftment outcomes between fresh or Microniche-expanded cells (Supplementary Fig. 2b, f). Together, these results indicated that the Microniche system could be used to expand functionally defined human HSCs.

As an HSC scarcity disease, aplastic anemia (AA) can serve as an informative model for assessing HSC function and quantity, as well as their ability to regenerate[34,35]. We therefore applied the Microniche system in AA HSCs culture and found that the proportion and absolute numbers of CD34+CD38− and CD34+CD38−CD49f+ cells both dramatically increased when starting from BM MNCs derived from AA patients compared with expansion in control medium (Fig. 1g, Supplementary Fig. 1h and Supplementary Table 1). Since it is well-known that HSCs in AA are unable to contribute to ongoing hematopoiesis[15,34], we designed xenograft assays to investigate the capacity for hematopoietic reconstruction by Microniche-derived AA HSCs. Our data showed that 3/7 recipients in the Microniche group exhibited successful human haematopoiesis, whereas no human haematopoiesis was detected in control group recipient mice (Fig. 1h). These findings suggested that our Microniche system can expand human HSCs from different sources in vitro and ex vivo while maintaining the stemness of the expanded cells.

### The Microniche culture system expands functional Mk-biased HSCs

Loss of the Mk lineage is a unique characteristic of AA patients, and the restoration of megakaryopoiesis, along with improved platelet counts, are still unmet clinical needs[34,35]. Since our results showed that Microniche expansion could promote the engraftment of residual HSCs from AA bone marrow, we hypothesized that the Microniche system could also expand functional Mk stem and/or progenitor cells. Microniche-based culture of mouse HSCs resulted in significantly higher proportions and absolute cell numbers of mouse phenotype-defined long-term HSCs (LT-HSCs) as well as Mk-biased HSCs compared with control cells cultured in conditioned

 

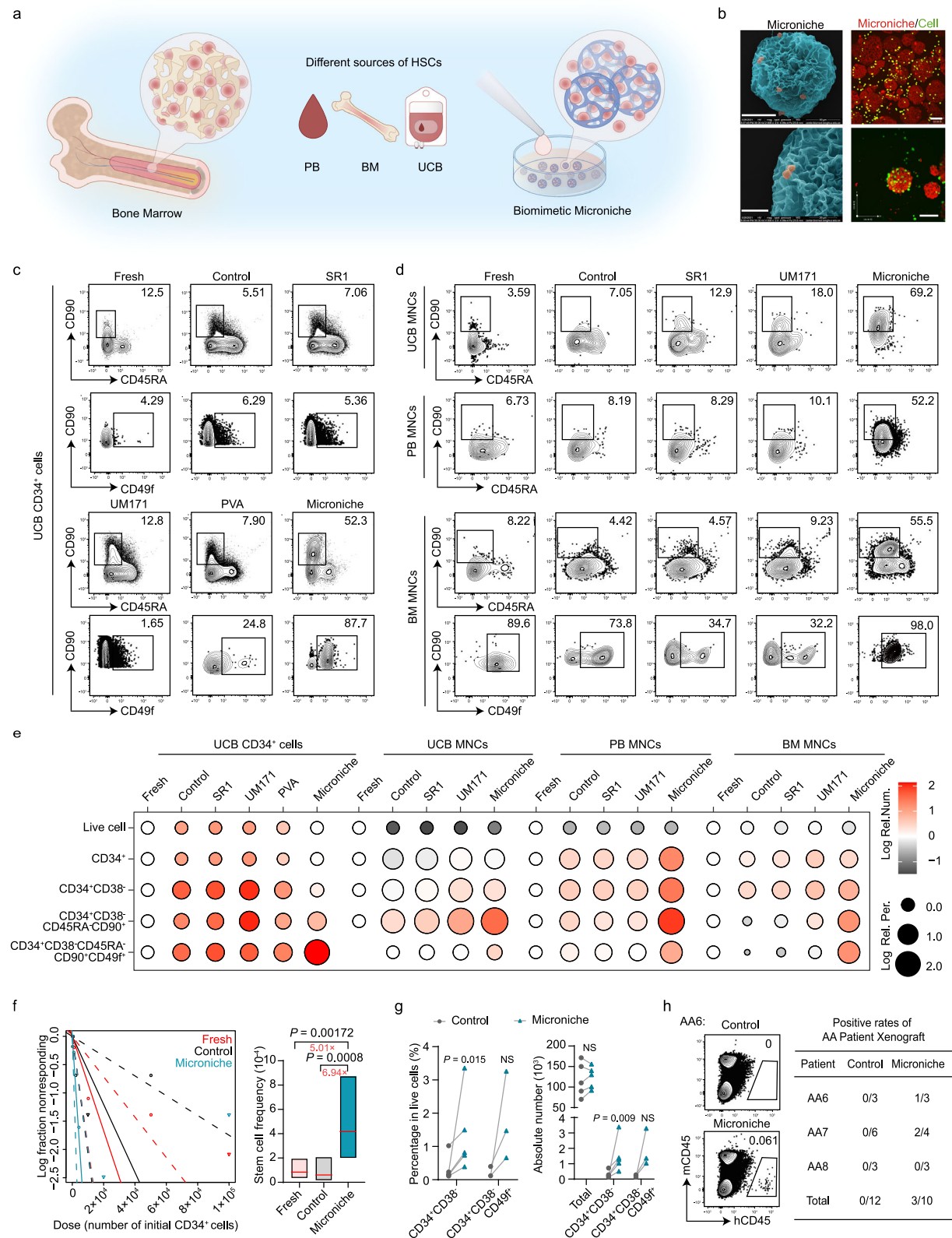

medium only (Supplementary Fig. 3a and Supplementary Table 2). Moreover, mouse Mk-CFU assays showed an increase in total Mk-colony numbers and larger colonies in the Microniche group relative to the control group (Supplementary Fig. 3b), indicating an improved ability to form colonies from mouse Mk progenitors. Interestingly, in human Mk-CFU assays, we found that cells from the Microniche-based system had lower total and large Mk-colony

counts in 1-week cultures compared to controls (Supplementary Fig. 3c), suggesting that this system could not promote the ability to form colonies in human Mk progenitors.

The above data led us to more closely examine whether the Microniche system could indeed increase the Mk lineage subpopulation through Mk-biased HSC expansion. Since human Mk-biased HSCs lack a clear immune phenotype, we examined the phenotype-defined

**Fig. 1 | The expansion of human HSCs in a Microniche-based culture system.**
**a** Schematic of HSC expansion using the biomimetic Microniche system in petri dishes. **b** Scanning electron microscope images of UCB CD34[+] cells cultured with Microniche (left). Scale bar, upper panel: 50 μm; lower panel: 20 μm. Representative images of co-immunofluorescent staining of live cells (Calcein AM, green) and Microniche (PI, red) in UCB CD34[+] cells (right). Scale bar, 100 μm. The images represent a mean of three independent experiments each done in six replicates. Representative FACS profiles of CD34[+]CD38[−]CD45RA[−]CD90[+] and CD34[+]CD38[−]CD45RA[−]CD90[+]CD49f[+] populations in UCB CD34[+] cells (**c**), UCB, PB or BM MNCs (**d**) before (fresh) and after culture in control medium (StemSpan SFE-MII + 1% P/S + 10 ng/ml hSCF + 100 ng/ml hTPO) or control medium supplemented with SR1 (1 μM), UM171 (40 nM), PVA (0.1%) or Microniche. For UCB CD34[+] cells, fresh n = 3 technical replicates; Microniche n = 3 biological replicates; other groups, n = 4 biological replicates. For BM and PB MNCs, n = 3 biological replicates; for UCB MNCs, fresh n = 4 technical replicates and other groups n = 3 biological replicates. **e** Dot plot showing the Log absolute cell counts (color) and Log percentages (size) of phenotype-defined cell subsets relative to the fresh cell group in the in vitro

culture assays of (**c**) and (**d**). Data were normalized with data of fresh group from each source. In UCB MNCs, the data of CD34[+]CD38[−]CD45RA[−]CD90[+]CD49f[+] was normalized with data of control group. **f** Graph of limiting dilution assays and stem cell frequency in primary engraftment. Apparent flow cytometric clustering of human CD45 was defined as successful engraftment. Fresh n = 43, Control n = 30, Microniche n = 36 biologically independent animals. Chi-square test. Solid lines indicate best-fit linear model and dashed lines confidence intervals. Box plots show the median (middle line) with the 25th and 75th percentiles (box). **g** Percentage and absolute number of total live cells, CD34[+]CD38[−] (n = 5 biological replicates) and CD34[+]CD38[−]CD49f[+] (n = 3 biological replicates) cells derived from AA patients after culture. Each pair of data points represents a single AA case. Paired two-tailed Student's t test. NS not significant. **h** Representative FACS profiles of the engrafting human CD45[+] cells in BM of NOG mice 16 weeks post-transplantation (left). Apparent flow cytometric clustering of human CD45 was defined as successful engraftment. Positive xenograft rates of control- and Microniche-expanded AA patient-derived MNCs (right).

HSCs and Mk progenitors (progeny of Mk-biased HSCs) in long-term culture assays and found that the proportions of CD34[+]CD38[−]CD45RA[−]CD90[+] and CD34[+]CD38[−]CD45RA[−]CD90[+]CD49f[+] HSCs were higher in the Microniche group than in the control and UM171 group after 4 weeks of culture. Notably, the CD34[+]CD38[−]CD45RA[−]CD90[+]CD49f[+] population plateaued at week 4. While the Mk progenitors (CD34[+]CD38[+]CD45RA[−]CD71[+]CD110[+] and CD34[+]CD38[+]CD41a[+] cells[15,36]) reached their apex at week 6 (Fig. 2a, b, Supplementary Fig. 3d and Supplementary Dataset 3). These results indicated that the Microniche-based strategy could expand Mk-biased HSC in a long-term culture system.

To accurately characterize Mk reconstruction in the xenograft model, we developed a flow cytometry gating strategy to avoid false positives in analysis of LDA-transplantation recipients caused by mouse Mk lineage cells. In this strategy, successful Mk reconstruction was indicated by sorting differentiated Mk lineage populations (i.e., distinguishing hCD41a[+], hCD41a[+]hCD61[+]hCD42b[−], or hCD41a[+]hCD61[+]hCD42b[+] [36,37] from mCD45[−]hCD45[−]mCD41[−]mCD42d[−]mCD62[−] cells) obtained from BM of recipient mice. In addition, successful Mk reconstruction was also determined by successful sorting of Mk progenitors (i.e., distinguishing hCD41a[+], hCD71[+]hCD110[+], or hCD71[−]hCD110[+] [15] from mCD45[−]hCD45[+]hCD34[+]hCD38[+] cells) in the BM of recipient mice (Fig. 2c).

Interestingly, in the fresh, control, and Microniche groups, the myeloid (hCD45[+]CD33[+]) and lymphoid (hCD45[+]CD19[+]) lineages were successfully reconstructed in most of the hCD45[+] recipients, whereas Mk lineage (hCD41a[+]) reconstruction was observed in only half or fewer of the engrafted recipients (Fig. 2d–f and Supplementary Dataset 2 and 4). These data supported that an HSC subpopulation−Mk-biased HSCs−was responsible for Mk reconstruction, which was less abundant than cells capable of myeloid or lymphoid reconstruction in this model. By contrast, human Mk progenitors were undetected, while differentiated Mk lineage cells were found in only one recipient in the control group, although these populations were present in more than 1/3 of the fresh cell and Microniche recipients (Fig. 2d–g and Supplementary Dataset 2 and 4), suggesting that Mk-biased HSCs, which are typically lost under culture conditions, were maintained by the Microniche system. Extreme limiting dilution analysis (ELDA) revealed a 3.36-fold expansion in Mk-biased HSCs in the Microniche group (Fig. 2h and Supplementary Dataset 2 and 4). Similar results were observed when hCD41a[+]hCD61[+]hCD42b[−] (2.85-fold) or hCD41a[+]hCD61[+]hCD42b[+] (2.75-fold) populations were used as the basis (Supplementary Fig. 3e, f and Supplementary Dataset 2 and 4). Taken together, our data demonstrated that Microniche could successfully expand functional Mk-biased HSCs.

## The Microniche-based culture system can support scalable HSC expansion from UCB mononuclear cells

We next attempted a large-scale expansion of UCB HSCs using a Microniche-based system in a stirred-tank bioreactor (Fig. 3a). UCB MNCs were directly cultured in the bioreactor with the Microniche. Photomicrographs of Wright-Giemsa stained MNCs from fresh and Microniche groups showed that cells of Microniche group were homogeneously round, with scant and agranular cytoplasm and round eccentric nuclei, while cells of fresh group resulted in heterogeneous populations of large cells with granular cytoplasm and irregular nuclei, indicating some degree of cell differentiation (Fig. 3b). These data indicated the Microniche-based system expands primitive HSCs. Flow cytometry analysis of phenotype-defined HSC expansion showed that the CD34[+]CD38[−]CD45RA[−]CD90[+]CD49f[+] populations reached the apex at day 3; the absolute number of HSPCs were highest in the 40 rpm stir speed group and the most suitable concentration for the Microniche was 50 mg per bottle (Supplementary Fig. 4a–f and Supplementary Table 3). In the most suitable condition (40-rpm stirring speed, a 50 mg Microniche per bottle, 3-day culture), the HSPC populations CD34[+], CD34[+]CD38[−], CD34[+]CD38[−]CD45RA[−]CD90[+], and CD34[+]CD38[−]CD45RA[−]CD90[+]CD49f[+] cells were expanded 6.2-, 6.5-, 20.8-, and 179-fold by the Microniche system, respectively (Fig. 3b, c and Supplementary Table 4). These results supported the scalability of Microniche-based expansion of UCB HSCs in vitro.

Limiting dilution xenograft assays in immunodeficient NCG mice were used to validate the expansion of functional HSCs. CD34[+] cells for fresh group and total cultured cells for Microniche group derived from equal initiated UCB MNCs were used in the dose dependent xenograftment (Supplementary Fig. 4g). ELDA analysis showed that Microniche-based scalable expansion resulted in a significantly greater frequency of human HSCs (7.39-fold) compared to that observed in xenografts of fresh cells, respectively (Fig. 3d, Supplementary Fig. 4h and Supplementary Dataset 5). In addition, Microniche-derived cells could differentiate into multiple lineages and retained their capacity for HSPCs reconstruction in vivo (Supplementary Fig. 4i, j), indicating they were unaffected by pre-graft expansion. Using the hCD41a[+] population from mCD45[−]hCD45[−]mCD41[−]mCD42d[−]mCD62[−] cells as a basis for Mk reconstruction, ELDA analysis revealed that the frequency of Mk-biased HSCs in Microniche group was 5.67-fold larger than that of fresh cells (Fig. 3e, Supplementary Fig. 4h and Supplementary Dataset 5). In addition, the Mk progenitors were also detectable in the recipients revived scalable expanded cells by Microniche-based system (Supplementary Fig. 4k). Together, these results indicated that the Microniche-based culture system can support scalable expansion of HSCs including Mk-biased HSCs using UCB MNCs as feedstock.

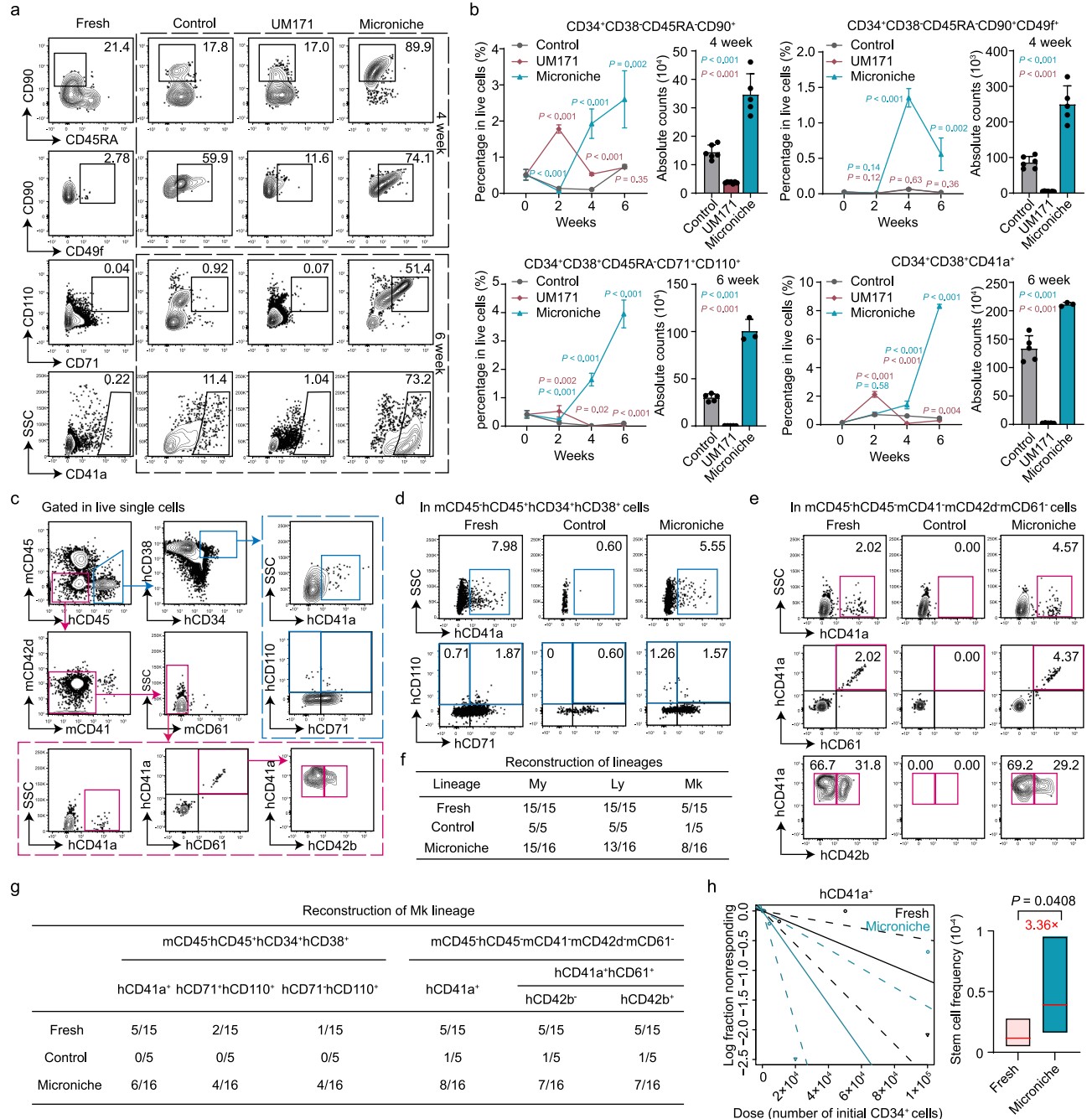

**Fig. 2 | Functional Mk-biased HSC expansion using the Microniche culture system in vitro and ex vivo.** Representative FACS profiles (**a**) and quantification (**b**) of phenotype-defined HSC and Mk subpopulations in long-term in vitro culture of UCB CD34⁺ cells. Percentage of the indicated cells relative to total live cells at weeks 0 to 6, and absolute cell counts at peak population density. Fresh n = 4 technical replicates; Control, week 2 and week 6 n = 5 biological replicates, week 4 n = 6 biological replicates; UM171, week 2 and week 6 n = 5 biological replicates, week 4 n = 7 biological replicates; Microniche, week 2 and week 4 n = 5 biological replicates, week 6 n = 3 biological replicates. All data represent means ± s.d.; comparisons with control by unpaired two-tailed Student's t test. **c**–**h** Analysis of Mk reconstruction in LDA xenografts. **c** Flow cytometry gating for human Mk lineage in BM cells of recipient NOG mice. Representative FACS profiles of Mk reconstruction in BM mCD45⁻hCD45⁺hCD34⁺hCD38⁺ (**d**) or mCD45⁻hCD45⁻mCD41⁻mCD42d

⁻mCD61⁻ (**e**) cells of NOG mice at 16 weeks post-transplantation. **f** The proportions of successfully reconstituted myeloid (My; mCD45⁻hCD45⁺hCD33⁺), lymphoid (Ly; mCD45⁻hCD45⁺hCD19⁺), and megakaryocytic (Mk; mCD45⁻hCD45⁻mCD41⁻mCD42d⁻mCD61⁻hCD41a⁺) lineages in hCD45⁺ engrafted mouse recipients. **g** The proportion of engrafted NOG mice with successful Mk lineage reconstruction based on the Mk population presence in BM mCD45⁻hCD45⁺hCD34⁺hCD38⁺ or mCD45⁻hCD45⁻mCD41⁻mCD42d⁻mCD61⁻ cell populations. **h** Mk-HSC frequency based on the mCD45⁻hCD45⁻mCD41⁻mCD42d⁻mCD61⁻hCD41a⁺ subpopulation 16 weeks post-transplantation. Fresh n = 43, Control n = 30, Microniche n = 36 biologically independent animals. Chi-square test. Solid lines indicate best-fit linear model and dashed lines confidence intervals. Box plots show the median (middle line) with the 25th and 75th percentiles (box).

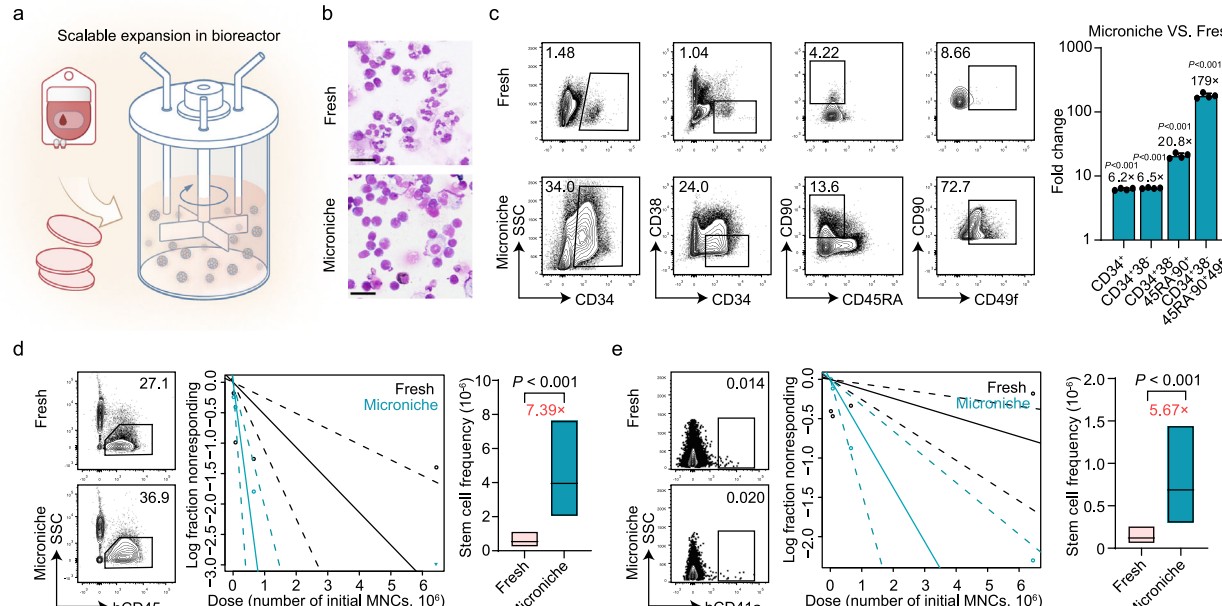

**Fig. 3 | Large-scale dynamic bulk-culture with Microniche. a** Schematic of HSC expansion using the biomimetic Microniche system in a large-scale, stirred bioreactor. **b** Photomicrographs of Wright-Giemsa stained fresh UCB MNCs and cells after culture in a dynamic stirred Microniche-based bioreactor. Scale bar, 25 μm. *n* = 3 biological replicates, with similar results. **c** Representative FACS profiles and fold change of absolute counts of indicated UCB MNC subpopulations before (fresh) and after 3 days of expansion in a dynamic stirred Microniche-based bioreactor. Fresh, *n* = 4 technical replicates; Microniche, *n* = 4 biological replicates. All data represent means ± s.d.; unpaired two-tailed Student's *t* test. **d** Representative FACS profiles and stem cell frequency based on the mCD45⁻hCD45⁺ subpopulation

16 weeks post-transplantation in xenograft NCG mice. Fresh *n* = 38, Microniche *n* = 40 biologically independent animals. Chi-square test. Solid lines indicate best-fit linear model and dashed lines confidence intervals. Box plots show the median (middle line) with the 25th and 75th percentiles (box). **e** Representative FACS profiles and Mk-HSC frequency based on the mCD45⁻hCD45⁻mCD41⁻mCD42d⁻mCD61⁻hCD41a⁺ subpopulation 16 weeks post-transplantation. Fresh *n* = 38, Microniche *n* = 40 biologically independent animals. Chi-square test. Solid lines indicate best-fit linear model and dashed lines confidence intervals. Box plots show the median (middle line) with the 25th and 75th percentiles (box).

## Mk-biased HSCs are enriched in the CD34⁺CD38⁻CD45RA⁻CD90⁺CD49fˡᵒʷCD62L⁻CD133⁺ subpopulation

To identify the immune-phenotype of human Mk-biased HSCs, we focused on differences in the immune-phenotypes between cells capable (fresh and Microniche groups) and incapable (control group) of Mk reconstruction. Flow cytometry analysis of CD34⁺ UCB cells and BM MNCs revealed that CD49f cells could be divided into CD49fⁿᵉᵍ, CD49fˡᵒʷ, and CD49fʰⁱᵍʰ populations. Specifically, amount of the CD49f positive cells were distributed in the CD49fˡᵒʷ gate for fresh and Microniche samples, whereas CD49f positive cells in the control group were primarily sorted into the CD49fʰⁱᵍʰ phenotype, and especially so in BM-derived samples (Supplementary Fig. 5a). Then, we sorted 200 cells each with CD49fˡᵒʷ or non-CD49fˡᵒʷ (CD49fⁿᵉᵍ&ʰⁱᵍʰ) phenotypes from the CD34⁺CD38⁻CD45RA⁻CD90⁺ population of fresh and Microniche groups, respectively, then used these cells for xenografts (Supplementary Fig. 5b). We found that a larger proportion of CD49fˡᵒʷ recipients reconstructed the Mk lineage than that of non-CD49fˡᵒʷ recipients, in both the fresh (4/7 vs. 2/12) and Microniche (7/9 vs. 4/7) groups (Supplementary Fig. 5c). Moreover, recipients of CD49fˡᵒʷ cells had significantly more platelets (PLTs) in PB at week 16 (Supplementary Fig. 5d). These data suggested that Mk-biased HSCs were enriched in the CD49fˡᵒʷ cell population in both the physiological state (i.e., fresh cells) and Microniche culture system.

We also sorted HSCs (CD34⁺CD38⁻CD45RA⁻CD90⁺) from the control and Microniche groups for analysis by BD Rhapsody single-cell RNA-seq with the oligonucleotide-conjugated antibodies against CD49f. The HSCs were sorted into CD49fⁿᵉᵍ (C1), CD49fˡᵒʷ (C2), and CD49fʰⁱᵍʰ (C3 and C4) clusters (Fig. 4a, b). We found that 95% of C2 and 53% of C3 cells were derived from the Microniche group, while 88% of C4 and 82% of C1 cells were derived from the control group (Fig. 4a). Interestingly,

long-term stem cell markers were enriched in the C2; Mk progenitor markers[12,37] were largely enriched in C3; and Mk lineage markers[12,37] were enriched in C3 and C4 (Fig. 4c). Trajectory and pseudotime analysis showed that C2 cells reside at the top of haematopoiesis hierarchy (Supplementary Fig. 5e). We found two surface marker genes, *SELL* (CD62L) and *PROM1* (CD133), among the top 10 genes in C2 (Fig. 4d). Flow cytometry targeting these markers revealed a specific CD34⁺CD38⁻CD45RA⁻CD90⁺CD49fˡᵒʷCD62L⁻CD133⁺ subpopulation present in both fresh and Microniche cells, but almost completely absent in control, SR1, UM171 and PVA cells (Fig. 4e), most prominently in BM-derived samples (Supplementary Fig. 5f). Moreover, Microniche-based scalable expansion in bioreactor also expanded this subpopulation for 123-fold in absolute counts compared to fresh cells (Supplementary Fig. 5g and Supplementary Table 4). These data indicated that the CD34⁺CD38⁻CD45RA⁻CD90⁺CD49fˡᵒʷCD62L⁻CD133⁺ subpopulation might be with the capacity of Mk lineage reconstruction.

Then we sorted the CD62L⁻CD133⁺ and non-CD62L⁻CD133⁺ cells for LDA xenograft assays from CD34⁺CD38⁻CD45RA⁻CD90⁺CD49fˡᵒʷ population (Supplementary Fig. 5h). We use no less than 0.01% hCD41a⁺ cells in mCD45⁻hCD45⁻mCD41⁻mCD42d⁻mCD62⁻ population of recipient BM cells at week 16 as a standard of successful Mk reconstruction and no less than 0.01% mCD45⁻hCD45⁺ cells as a standard of successful reconstruction for My and Ly lineages. Our data showed that 15/55 recipients of CD62L⁻CD133⁺ group, and 3/50 of the non-CD62L⁻CD133⁺ group successfully reconstructed human Mk lineage (Fig. 4f). For each different cell dose of injection, recipients of CD62L⁻CD133⁺ group exhibited higher average percentages hCD41a cells in BM than those of non-CD62L⁻CD133⁺ group (Supplementary Fig. 5i). ELDA analysis revealed that the frequency of Mk-biased HSCs in CD62L⁻CD133⁺ cells was 3.19-fold larger than that of

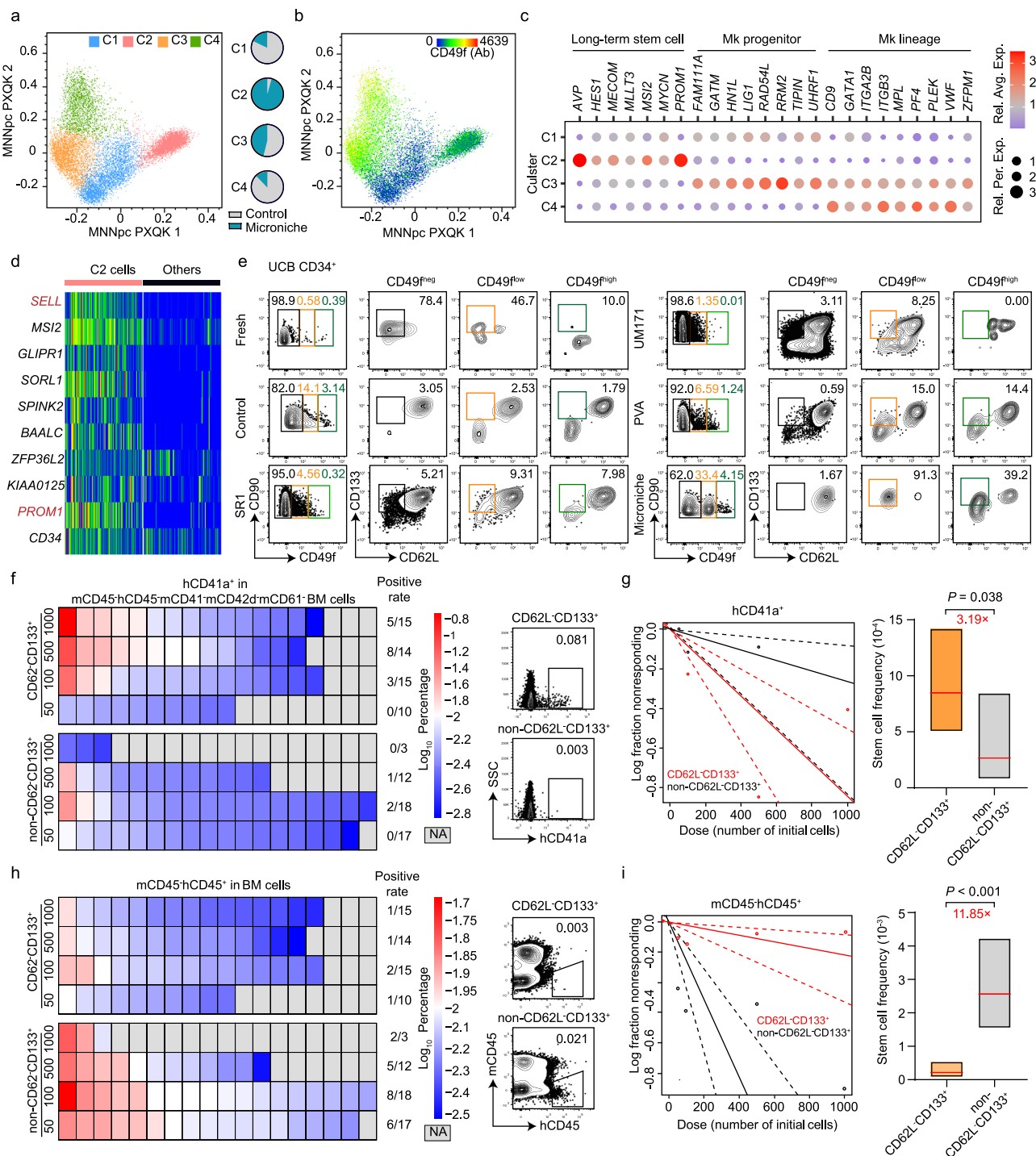

non-CD62L⁻CD133⁺ cells (Fig. 4g). On the contrary, hCD45⁺ reconstruction could be observed in 5/55 recipients of CD62L⁻CD133⁺ group, but 21/50 recipients in non-CD62L⁻CD133⁺ group (Fig. 4h). In each inner-dose comparison, recipients of non-CD62L⁻CD133⁺ group had a significantly higher level of hCD45 than CD62L⁻CD133⁺ group (Supplementary Fig. 5j). And ELDA analysis revealed an 11.85-fold of stem cell frequency in non-CD62L⁻CD133⁺ group compared to CD62L⁻CD133⁺ group (Fig. 4i). And in cell dose 500 and 100, recipients of CD62L⁻CD133⁺ group had significantly more PLTs in PB than recipients of non-CD62L⁻CD133⁺ group (Supplementary Fig. 5k). These data suggested that functional human Mk-biased HSCs were enriched in the CD34⁺CD38⁻CD45RA⁻CD90⁺CD49f^low CD62L⁻CD133⁺ cell population.

## Cytokine milieu affects microniche-based expansion of Mk-biased HSCs

To clarify the mechanism by which the Microniche system mediates HSC expansion, we used RNA-seq to identify transcriptomic differences between the Microniche-expanded and control cells. A total of 273 differentially up-regulated genes, but only 5 down-regulated, in the Microniche cultured cells (Supplementary Fig. 6a). Gene ontology (GO) and Kyoto Encyclopedia of Genes and Genomes (KEGG) analysis revealed significant enrichment for several gene sets and pathways in the Microniche group (Supplementary Fig. 6b, c), among which we found that chemokine- and cytokine-related pathways were highly significant (Fig. 5a). Using the reported gene sets (see "Methods") and the high-expressed genes in cluster 2 in Fig. 4a as Mk-biased HSC gene

**Fig. 4 | Determination of functional Mk-biased HSC immune phenotype. a–d** BD Rhapsody single-cell RNA-seq profiles of CD34⁺CD38⁻CD45RA⁻CD90⁺ cells sorted after culture in Microniche or control medium. **a** Four cell clusters were identified by k-means clustering analysis. Each dot represents one cell and colors represent distinct cell clusters (C1–C4), as indicated. Pie chart showing the percentages of each cell cluster in control and Microniche groups. **b** CD49f expression in each cell cluster. Ab refers to antibody. **c** Expression of the indicated genes in the four distinct clusters. Bubble color indicates Log relative average expression; bubble size indicates Log relative percentage expression. **d** Heatmap of scaled expression for top 10 highest differentially expressed genes (DEGs) in the C2 cluster compared to other clusters, including surface marker genes *SELL* (also known as CD62L) and *PROM1* (also known as CD133). **e** Representative FACS profiles of CD62L⁻CD133⁺ distribution within CD34⁺CD38⁻CD45RA⁻CD90⁺CD49f^{neg/low/high} subpopulations of UCB CD34⁺ cells before (fresh) and after 7 days of culture in control medium or control medium supplemented with SR1 (1 μM), UM171 (40 nM), PVA (0.1%) or

Microniche. **f** Heatmap of Log percentage (left) and representative FACS profiles (right) of hCD41a⁺ cells in mCD45⁻hCD45⁻mCD41⁻mCD42d⁻mCD61⁻ population of recipient BM cells. Positive xenograft rates in groups were listed. **g** Mk-HSC frequency based on the mCD45⁻hCD45⁻mCD41⁻mCD42d⁻mCD61⁻hCD41a⁺ population 16 weeks post-transplantation. CD62L⁻CD133⁺ group $n = 54$, non-CD62L⁻CD133⁺ group $n = 50$ biologically independent animals. Chi-square test. Solid lines indicate best-fit linear model and dashed lines confidence intervals. Box plots show the median (middle line) with the 25th and 75th percentiles (box). **h** Heatmap of Log percentage (left) and representative FACS profiles (right) of mCD45⁻hCD45⁺ cells in recipient BM cells. Positive xenograft rates in groups were listed. **i** HSC frequency in xenograft NOG mice 16 weeks post-transplantation. CD62L⁻CD133⁺ group $n = 54$, non-CD62L⁻CD133⁺ group $n = 50$ biologically independent animals. Chi-square test. Solid lines indicate best-fit linear model and dashed lines confidence intervals. Box plots show the median (middle line) with the 25th and 75th percentiles (box).

set, we then used Gene Set Enrichment Analysis (GSEA) to evaluate the enrichment of cytokine- and stemness-related gene sets (quiescence, cell arrest) as well as Mk-biased HSC genes. All 4 gene sets were enriched in the Microniche group (Fig. 5b and Supplementary Fig. 6d). Real-time PCR confirmed the up-regulation of most genes in the cytokine-cytokine receptor pathway under Microniche culture conditions (Fig. 5c and Supplementary Table 5). Luminex liquid suspension chip detection was used to compare differences between the cytokine profiles secreted in Microniche and control culture medium. This analysis showed that for the majority of detected cytokines, the concentrations were higher in the Microniche group than that in control group (Fig. 4d and Supplementary Table 6). These results suggested that the cytokine milieu was recapitulated in the Microniche-based culture microenvironment.

Next, 10X Genomic scRNA-seq showed that CD34⁺ cells from the control and Microniche groups were sorted into nine clusters (Fig. 5e and Supplementary Fig. 6e, f). We found higher proportions of cells derived from the Microniche group than from the control group in Clusters 1, 6, and 8 (C1, C6, and C8) (Fig. 5e). The C6 group was then designated as Mk and erythroid (Er) stem and progenitor cells based on enrichment for Mk and Er lineage marker genes such as *ITGA2B* (CD41), *VWF*, *ITGB3* (CD61), *NFE2*, *GATA1*, and *TFRC* (CD71)[12,15,37], as well as enrichment for stem cell marker genes *TAL1*, *IGTA6* (CD49f), and *HES1*[33,38] (Fig. 5f). Interestingly, we found that most of the cytokine genes illustrated in Fig. 5c, d were expressed in C1 (Fig. 5f), 84% of which were derived from the Microniche group (Fig. 5e). Moreover, we investigated inter-cluster cytokine-receptor communication using CellPhoneDB[39] and found that C1 communicated broadly with other clusters (Fig. 5g). These data indicated that the specific cell population (C1) expanded by Microniche culture showed a profile of secreted cytokines that could potentially contribute to the expansion of Mk-biased HSCs. These data indicated that the specific cell population (C1) expanded by Microniche showed a profile of secreted cytokines that could potentially contribute to the expansion of Mk-biased HSCs. Furthermore, we found that addition of 5 cytokines high-frequently appeared in our data resulted in greater numbers of CD49f^{low} and CD62L⁻CD133⁺ populations compared with unsupplemented cultures (Fig. 5h, i and Supplementary Table 7). These data suggested that the cytokine milieu contributes to enhancing expansion of Mk-biased HSCs.

### Proper biomimicry of niche-like scaffold contributes to the expansion of Mk-biased HSCs

Considering that addition of 5 cytokines did not fully reproduce the effect of the Microniche in the expansion of primitive HSCs (Fig. 5h, i and Supplementary Table 7), we speculated that the biomimetic structure of the Microniche contributed to the expansion of primitive HSCs. To validate this hypothesis, we compared the proximity of the physical parameters of four different microcarriers with those of the

BM microenvironment (Fig. 6a and Supplementary Fig. 7a). In these microcarriers, the composition of Cytodex 3 and Cytopore 1 were dextran, and the composition of Microcarrier W01 and Microniche were gelatin (hydrolyzed collagen)[40,41]. Only Cytodex 3 was poreless while other three microcarriers were porous with an average pore size of 20–30 μm, which was similar with that of BM (5-100 μm)[40,41]. The average Young's Modulus of Microniche was 22 kPa, which was closer in approximation to BM (0.3-24 kPa)[42–44] than the other three microcarriers (40 kPa).

Through flow cytometry, we evaluated the expansion of five HSPC populations (P1: CD34⁺, P2: CD34⁺CD38⁻, P3: CD34⁺CD38⁻CD45RA⁻CD90⁺, P4: CD34⁺CD38⁻CD45RA⁻CD90⁺CD49f^{low}, and P5: CD34⁺CD38⁻CD45RA⁻CD90⁺CD49f^{low}CD62L⁻CD133⁺) by these four microcarriers and gelatin supplemented conditional medium, respectively. Our data show that Cytodex 3, Cytopore 1, and Microcarrier W01 failed to expand any HSPC population (Figs. 5b and 6c, Supplementary Fig. 7b and Supplementary Dataset 6). Cultures in the control medium, supplement of gelatin, and supplement of Microniche were able to expand these HSPC populations (Fig. 6b, c, Supplementary Fig. 7b and Supplementary Dataset 6). Microniche cultures had the highest cell counts of both primitive (P4, 3.91-fold) and Mk-biased HSCs (P5, 28.85-fold), while culture in conditional medium with or without gelatin had the same level of expansion for these two populations (P4: 2.69-, 1.11- fold; P5: 3.09-, 3.52-fold). These data demonstrated that the properly simulating of BM niche's physical structure by the Microniche was essential for expansion of primitive HSCs, especially for Mk-biased HSCs.

Recent studies have reported that Mks with distinct expression signatures are spatially organized in the BM niche[45]. Thus, we speculated that Mk-biased HSCs tended to be located in niche-like microenvironments. We then assessed the proportion of HSPCs inside and outside of the Microniche after culturing. Our data showed that about 20% total cells were located inside of Microniche, and that the cell proportions of P1 and P2 were higher in the outside than that in the inside (Fig. 6d, e). The cell proportions in P3 had no significant difference between inside and outside, whereas P4 and P5 cell proportions were higher in the inside. When we normalized the cell proportion with live cell proportion of each group, we found all these HSPCs populations were enriched inside the Microniche rather than outside (Fig. 6d, e). These data suggested that primitive and Mk-biased HSCs tended to locate inside of the Microniche during culturing.

Next, RNA-seq was used to identify transcriptomic differences between the inside and outside cells of the Microniche. We found a total of 70 differentially up-regulated genes and 23 down-regulated in the inside cells relative to the outside cells (Fig. 6f). GO and KEGG analysis also revealed highly significant enrichment of cell-matrix interactions and chemokine-related pathways (Fig. 6g and Supplementary Fig. 7c, d). Using the gene sets used in Fig. 5b, GSEA showed that cytokine-, stemness-, and Mk-biased HSC-related genes were

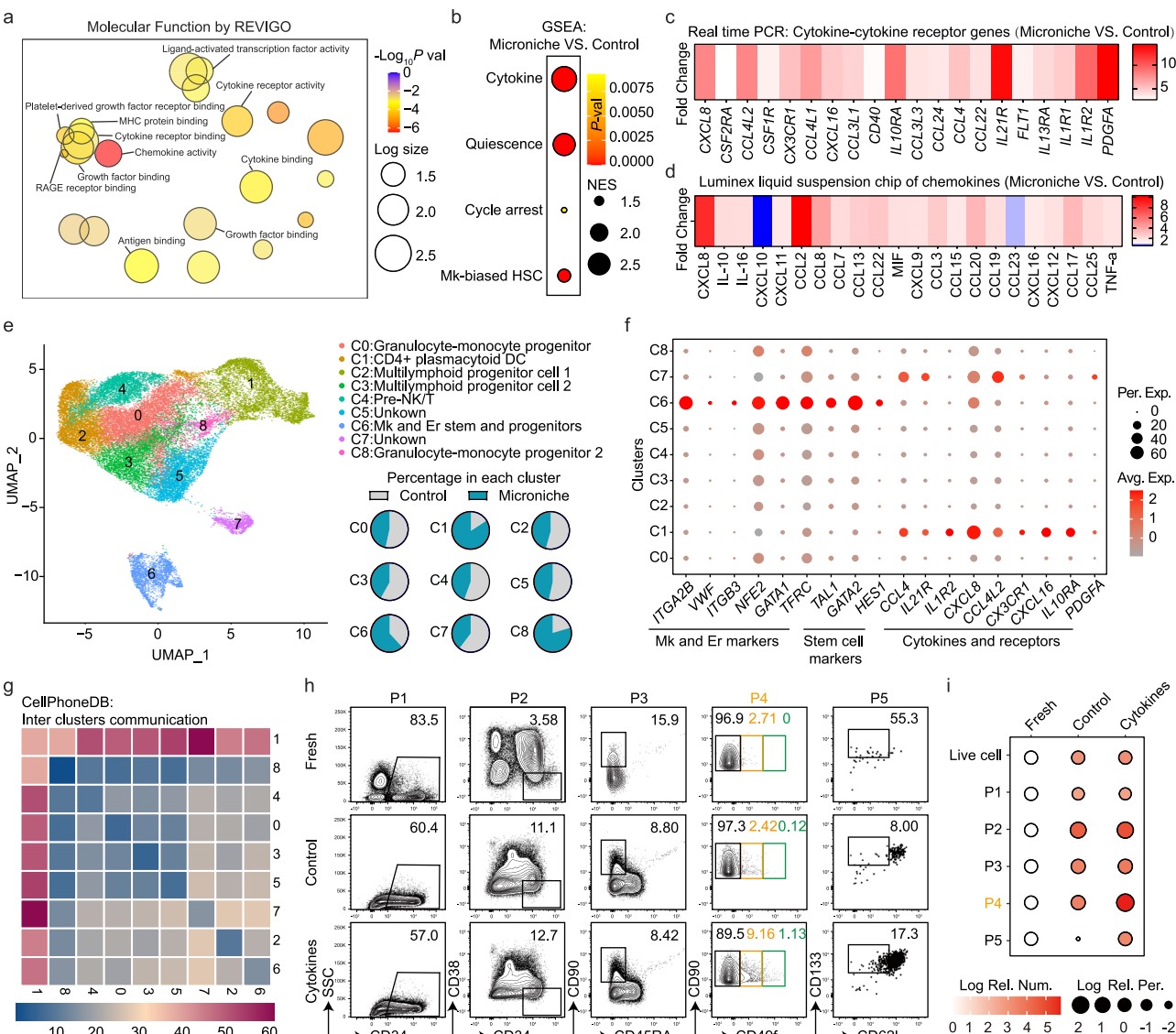

**Fig. 5 | Cytokine profiles between control medium and Microniche. a** REVIGO interactive graph of major Gene Ontology molecular function (GO MF) terms in RNA-seq of sorted CD34[+] cells after culture. Bubble color indicates $-\text{Log}_{10}$ $p$ value; bubble size indicates GO term frequency. **b** GSEA of the indicated gene sets enriched in Microniche vs. control. Bubble size indicates normalized enrichment score (NES); bubble color indicates $p$ value. **c** Heatmap of average relative expression for cytokine-cytokine receptor genes enriched in Microniche-cultured CD34[+] cells. Color indicates fold change in expression levels quantified by qPCR. **d** Heatmap showing the average relative secretion of chemokines enriched in Microniche culture supernatant. Color indicates fold change in chemokine levels determined by Luminex liquid suspension chip. **e** Identification of nine cell clusters visualized by UMAP in 10X Genomics single-cell RNA-seq profile of CD34[+] cells sorted after culture. Each dot represents one cell, and colors represent distinct cell clusters (C0–C8). Pie chart showing the percentages of each cell cluster in control and Microniche cultures. **f** Dot plot showing the expression of feature genes related to Mk and Er lineage, HSC stemness, chemokines, cytokines, and receptors in each cell cluster. Bubble color indicates average relative expression; bubble size indicates percentage of relative expression. **g** CellPhoneDB analysis of crosstalk between cell clusters. **h** Representative FACS profiles of phenotype-defined cell subsets in UCB CD34[+] cells before (fresh) and after cultured with or without cytokines (CXCL8, CCL24, IL-10, IL-1β, IL-1α) for 7 days. P1: CD34[+], P2: CD34[+]CD38[-], P3: CD34[+]CD38[-]CD45RA[-]CD90[+], P4: CD34[+]CD38[-]CD45RA[-]CD90[+]CD49f[low], P5: CD34[+]CD38[-]CD45RA[-]CD90[+]CD49f[low]CD62L[-]CD133[+]. **i** Dot plot showing the absolute numbers and percentages of phenotype-defined cell subpopulations. For each subpopulation, their changes after culture are summarized as Log number and Log percentage relative to fresh, with color (red) depth indicating the size of absolute numbers and bubble size indicating the size of percentages in living cells. Fresh, $n$ = 3 technical replicates; control and cytokines, $n$ = 4 biological replicates. Data were normalized with data of fresh group.

enriched in the inside cells (Fig. 6h and Supplementary Fig. 7e), suggesting that the niche-like physical structure contributed to the alteration of cytokine milieu by Microniche. It was reported that CXCR4[hi] monocytes are retained in the bone marrow until they switch to a proliferative state with increased expression of CCR2[46]. The expression of CXCR4 and its ligand CXCL12 (also known as stromal cell-derived factor-1, SDF-1) and CCR2 were further tested in both the inside and outside cells. The result showed that the expression of CXCL12 and CXCR4 were significantly higher in inside cells compared

to that of the outside cells, and the expression of CCR2 had no significant difference between these cells, suggesting that the CXCR4[hi] cells were retained inside of Microniche as in BM (Fig. 6i). We further found that addition of CXCL12 was not able to increase the primitive HSCs compared to culture using only control medium. The blocking of CXCR4 will significantly decrease the expansion effect of primitive HSCs by Microniche (Supplementary Fig. 8a, b). These data suggested that physical scaffolding was essential for the expansion of Mk-biased HSCs. Overall, these data supported that the proper biomimicry of

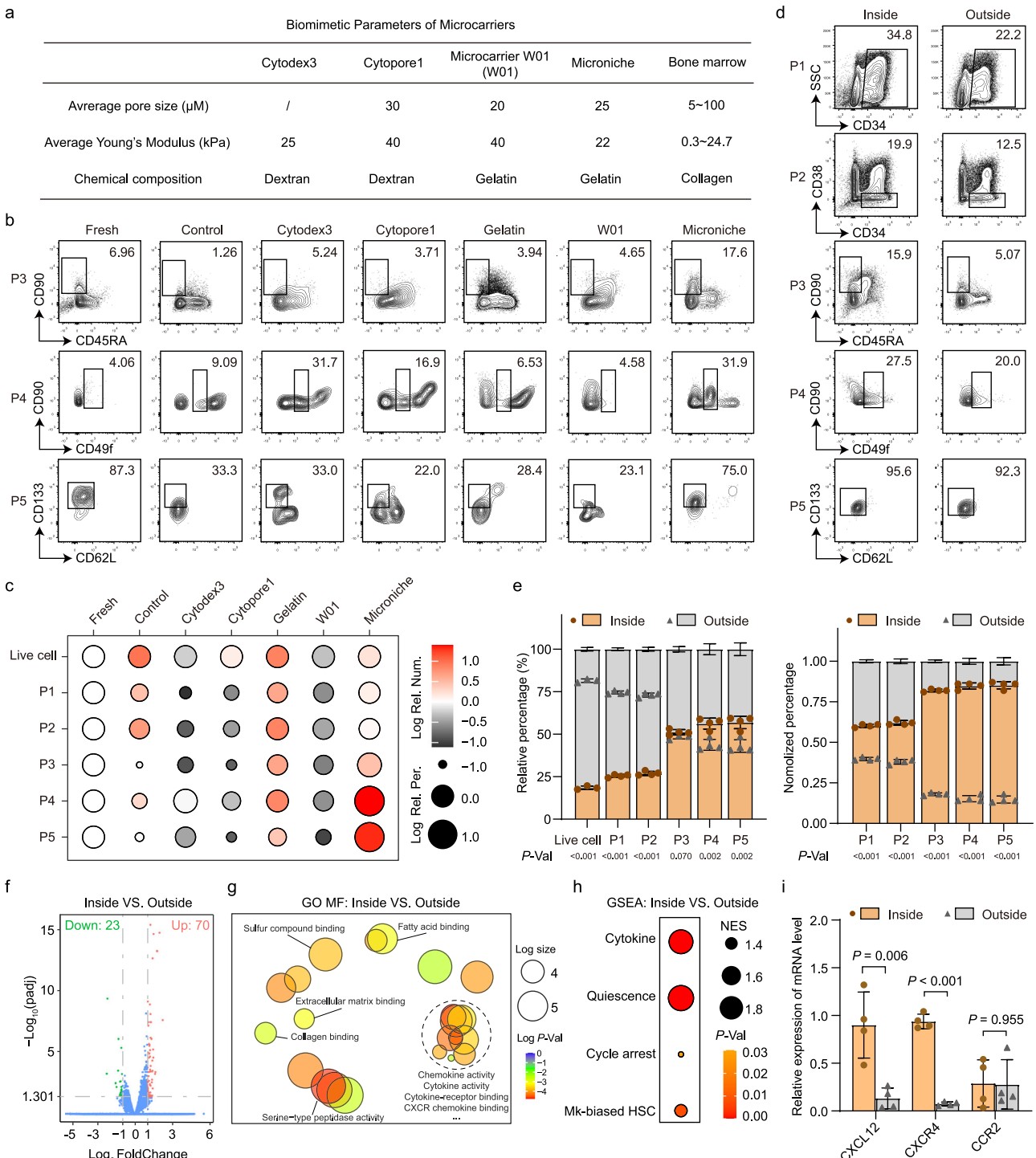

**Fig. 6 | The effect of physical structure to expansion of primitive and Mk-biased HSCs. a** Comparison of biomimetic parameters between different biomaterials and bone marrow. **b** Representative FACS profiles of CD34⁺CD38⁻CD45RA⁻CD90⁺ (P3), CD34⁺CD38⁻CD45RA⁻CD90⁺CD49f^low (P4), and CD34⁺CD38⁻CD45RA⁻CD90⁺CD49f^low CD62L⁻CD133⁺ (P5) populations in UCB CD34⁺ cells before (fresh) and after culture in control medium or control medium supplemented with Cytodex3, Cytopore1, Gelatin, Microcarrier W01, or Microniche. Fresh, *n* = 3 technical replicates; other groups, *n* = 3 biological replicates. **c** Dot plot showing the Log absolute cell counts (color) and Log percentages (size) of phenotype-defined cell subsets relative to the fresh cell group in the in vitro culture assays of (**b**). Data were normalized with data of fresh group. **d** Representative FACS profiles of CD34⁺ (P1), CD34⁺CD38⁻ (P2), P3, P4, and P5 populations inside or outside Microniche in cultured UCB CD34⁺ cells. **e** Relative percentage (left) and normalized percentage

(right) of live, P1, P2, P3, P4, and P5 populations inside or outside the Microniche after 7 days in culture (*n* = 4 biological replicates). All data represent means ± s.d. **f** Volcano plot of differentially expressed genes (DEGs) between cultured cells inside and outside the Microniche. **g** REVIGO interactive graph of major Gene Ontology molecular function (GO MF) terms in RNA-seq analysis of cultured cells inside and outside the Microniche. Bubble color indicates Log *p* value; bubble size indicates GO term frequency. **h** GSEA of the indicated gene sets enriched in cultured cells inside vs. outside Microniche. Bubble size indicates normalized enrichment score (NES); bubble color indicates *p* value. **i** Relative expression for the genes implicated in HSC maintenance and retention in BM between cultured cells inside and outside the Microniche (*n* = 4 biological replicates). All data represent means ± s.d. Unpaired two-tailed Student's *t* test.

Microniche contributed to the expansion of primitive HSCs and Mk-biased HSCs.

## Discussion

HSC transplantation (HSCT) still remains the only curative therapy for a number of hematological malignancies, such as leukemia and lymphoma, as well as non-malignant blood disorders such as immunodeficiency diseases, autoimmune conditions, hereditary blood disorders and anemias[47]. In addition to low quantity, the "loss of function" of HSC restricts the use of HSCT. Clinical trials of HSCT after UCB HSC ex vivo culture with agents showed that compared to the promotion of neutrophils reconstruction, the delay and failure of platelet recovery still remain a problem that needs to be addressed[48–50]. Recent studies have implied the existence of Mk-biased HSC in human hematological hierarchy[11–13,15], which may serve as a breakthrough to overcome platelet recovery-related problems faced in HSCT.

This work demonstrated the in vitro expansion of Mk-biased HSCs using a biomimetic Microniche-based culture system. Preliminary xenograft assays showed that the Mk lineage was almost lost following culturing in conditional medium supplemented with conventional factors, whereas My and Ly lineages were successfully restored upon transplantation of expanded UCB HSCs. These findings were consistent with that of cultured HSCs, which were hardly able to recover platelets in clinical trials[6]. However, the fresh UCB HSCs were able to reconstruct the Mk lineage in NOG mice, suggesting the presence of a previously unrecognized subpopulation capable of Mk reconstruction (i.e., Mk-biased HSCs) within the heterogeneous human HSC pool and the Mk-biased HSCs were sensitive to ex vivo culture. In this study, we found a special population of CD62⁻CD133⁺ were absence in the conventional HSC-expanding agent groups. These results were concurrent with the results showing that Mk haematopoiesis was sensitive to stimulation in both mice and humans[51]. Using our Microniche-based system resulted in an increase of successful Mk lineage reconstruction ex vivo. Thus, the reconstructive ability of the Mk lineage of HSC should be valued as an important indication in the "High-Fidelity" of HSC function evaluation, which has been ignored in previous studies[16–18,20–22].

Our data demonstrated Mk-biased HSCs were enriched in a CD49f^low population similar to long-term HSCs reported in a previous study[33]. ScRNA-seq analysis further revealed that the proportion of Microniche-expanded cells containing the Mk-biased HSCs specifically expressed markers of stemness rather than Mk lineage markers. Whereas HSCs expressing Mk markers in the medium control could not succeed in the reconstruction of Mk lineage, suggesting human Mk-biased HSCs were primitive cells. These findings were also consistent with previous works showing that Mk-biased HSCs reside at the apex of the HSC hierarchy, and that the Mk lineage is the predominant fate of native long-term hematopoietic stem cells in mice[11–13]. We found a special CD62⁻CD133⁺ population in human long-term HSCs was capable of Mk reconstruction, whereas the complementary non-CD62⁻CD133⁺ population was able to regenerate CD45⁺ cells. These data supported the heterogeneity of HSCs[10] and depicted a clearer figure of human Mk-biased HSCs. Using CD62⁻CD133⁺ as a phenotype marker base on the long-term HSC markers may improve the phenotyping of primitive HSCs and Mk-biased HSCs. After excluding the mouse Mk cells, most of the Mk reconstruction in the recipient mice were very low. On one aspect, the Mk lineage only take a very low proportion (about 0.05%) of MNCs in human BM. On another, species differences between human and mouse may be a reason of low Mk reconstruction. More strains of immune-deficient mice suitable for Mk reconstruction evaluation should be developed.

Recent studies have shown that the cytokines involved in immune regulation also participate in diverse functions of adult bone marrow Mks[30,52,53]. Similarly, we found high levels of some of the same cytokines and receptors (e.g., CXCL8, CCL3L1, CCL4, and IL10RA) in the Microniche culture medium, suggesting that culture in the Microniche may result in Mk-biased HSC expansion by sufficiently recapitulating the natural human BM cytokine milieu. Furthermore, the addition of five cytokines increased primitive HSC populations in conventional cultures, implying that cytokine signaling is essential for Mk-biased HSC expansion. Analysis of the physical parameters of four different microcarriers showed that the Microniche properly simulated the physical structure of the BM microenvironment. Based on its effective recapitulation of major structural, biophysical, and biochemical features simulating the bone marrow microenvironment, only the Microniche successfully maintained and expanded Mk-biased HSCs. Mk-biased HSCs are enriched in the pores of the Microniche, which is consistent with the report that Mks spatially located in BM niches[45]. The inside cells were also characterized with the same chemokine-receptors marker as the BM-located cells. These data emphasized the opinion addressed by previous work[22] that special scaffolding benefits in the fidelity of HSCs. That should be more closely studied in research related to HSC expansion.

More importantly, the Microniche developed in this study facilitated the expansion of HSCs in MNCs obtained from different sources, implying that the CD34⁺ cell sorting step in conventional methods could be omitted. This study also showed that BM, PB, and UCB MNCs could all serve as sources for HSC expansion using the Microniche system, which hinted at its considerable clinical application potential in expanding HSCs for transplantation[35,54]. In addition, our study showed that Microniche is scalable for in vitro expansion of one or more units of UCB MNCs in a stirred-tank bioreactor. The functionality of the produced cells was also supported by the xenograft assays, implying the widening applications for future UCB HSC studies. Collectively, by improving the standard of HSC function evaluation −"High-Fidelity" expansion with Mk recovery ability, demonstrating deep insight into human HSC heterogeneity−identification of human Mk-biased HSCs, and providing a flexible human HSC expansion strategy with considerable application potential, this study has important implications for both basic HSC research and prospective advances in clinical hematology.

## Methods

### Ethical issues

All primary cells followed the Declaration of Helsinki and were approved by the Ethics Review Board of the Institute of Hematology and Blood Diseases Hospital, Chinese Academy of Medical Sciences. Informed written consent was obtained from all donors according to humanitas ethical committee regulations from the Institute of Hematology and Blood Diseases Hospital (ethical review approval No: SBKT2020007-EC-2). The animal study was performed with the approval of the Animal Care and Use Committee of State Key Laboratory of Experimental Hematology, Institute of Hematology and Blood Diseases Hospital. All mouse experimental procedures were performed in accordance with the Regulations for the Administration of Affairs Concerning Experimental Animals approved by the State Council of the People's Republic of China.

### Human hematopoietic cells

Fresh umbilical cord blood and adult peripheral blood were obtained from consenting donors by Shandong Cord Blood Hematopoietic Stem Cell Bank, while bone marrow from healthy donors and aplastic anemia patients were obtained from Blood Biobank of Institute of Hematology and Blood Diseases Hospital (please refer to "Reporting summary" for details). Histopaque®-1077 (Sigma-Aldrich) was used to separate mononuclear cells. CD34⁺ cells were isolated using CD34 MicroBeads and a QuadroMACS Separator (Miltenyi Biotec) according to the manufacturer's instructions. Unless specified, each independent experiment used a mixture of no less than three units of umbilical cord blood as initial fresh cells for subsequent experiments to reduce individual variability.

## Mice

Female NOD.Cg-*Prkdc*scid*Il2rg*tm1Sug/JicCrl (NOG) mice (6–7 weeks) were purchased from Charles River Laboratories, female NOD/ShiLtJGpt-*Prkdc*em26Cd52*Il2rg*em26Cd22/Gpt (NCG) mice (6–7 weeks) were purchased from GemPharmatech Co., LTD and female C57BL/6J mice (8–10 weeks) were purchased from Beijing HFK Bioscience Co., LTD. All mice were housed under specific-pathogen-free (SPF) conditions, light/dark cycle: 12 h/12 h, temperature: 18–23 °C, humidity: 40–60%, with free access to food and water. Mice were randomly and evenly classified, by weight, into different experimental groups. At the end of all experiments, animals were euthanized under $CO_2$ anesthesia. Mouse BM cells were flushed out from ilia, femurs, and tibias into phosphate-buffered saline (PBS) with 2 mM ethylenediaminetetraacetic acid (EDTA, Sigma-Aldrich) for the follow-up experiments. BM c-Kit+ cells were isolated using mouse CD117 MicroBeads (Miltenyi Biotec, 130-091-224) according to the manufacturer's instructions.

## Atomic force microscopy (AFM)-applied elasticity measurement of microcarriers

The Young's Modulus of Cytopore1 (GE Healthcare, 17-0911-01), Cytodex3 (GE Healthcare, 17-0485-01), and two types of Microcarriers (CytoNiche, Beijing, Microcarriers-W01, Microniche)[25–28] were measured by AFM and compared with the elasticity of natural bone marrow (Using Yong Modulus in the range of 0.3–24.7 Kpa)[42–44]. The microcarriers were dispersed with Cell-Tak (Corning) and attached to a coverslip, while samples attached to the coverslip were mounted on an inverted fluorescence microscope (Zeiss Observer A1 stand) and measured using an AFM module CellHesion® 200 (JPK Instruments). The silicon tipless cantilever (ARROW-TL1-50, NANOWORLD) that was used to indent the samples had a nominal spring constant of 0.03 N/m and was attached with a silicon microsphere (20 μm in diameter) on the cutting edge. For each sample, AFM indentations at different locations ($n \geq 30$) of the cell were performed with -0.6 nN force at a 10 μm/s displacement rate. The effective Young's modulus was obtained by fitting the force-displacement curve to the Hertz/Sneddon model.

## In vitro culture and HSPCs harvest

For the freshly isolated CD34+ cells from UCB or MNCs from UCB, PB, and BM, $2 \times 10^5$ initial seeding cells/well (24-well plate; Corning, 3527) were cultured in a medium composed of StemSpan SFEMII (STEMCELL Technologies), 1% penicillin/streptomycin (P/S; Sigma-Aldrich), 10 ng/ml human stem cell factor (hSCF; PeproTech), and 100 ng/ml human thrombopoietin (hTPO; PeproTech). The cultures were performed in a 2 ml medium/well supplemented with or without 1 μM SR1 (Alichem, 41864), 40 nM UM171 (APExBIO, A8950), 0.1% PVA (Sigma, P8136-250G), 4 mg/well Cytopore1 (GE Healthcare, 17-0911-01), 10 mg/well Cytodex3 (GE Healthcare, 17-0485-01), 10 mg/well Gelatin (CytoNiche), 10 mg/well Microcarrier W01 (CytoNiche), or 10 mg/well Microniche (CytoNiche, F01-50). The cells were cultured at 37 °C with 5% $CO_2$ for 7 days.

For freshly isolated MNCs from BM of aplastic anemia patients, $1$–$3 \times 10^5$ initial seeding cells/well (specified in Supplementary Table 1) were cultured in a medium composed of Iscove's Modified Dulbecco's Medium (IMDM; Gibco), 10% fetal bovine serum (FBS; Gibco), 1% P/S, 100 ng/ml hSCF, 100 ng/ml hTPO, 100 ng/ml human fms-related tyrosine kinase 3 ligand (hFlt3L; PeproTech), 100 ng/ml (hIL-3; PeproTech), 100 ng/ml (hIL-6; PeproTech), and 100 ng/ml (hEPO; PeproTech). Cultures were performed in a 2 ml medium/well, supplemented with or without 5 mg/well Microniche at 37 °C with 5% $CO_2$ for 7–14 days.

For freshly isolated c-Kit+ cells from C57BL/6J mice, $2 \times 10^5$ initial seeding cells/well were cultured in a medium composed of IMDM, 15% FBS, 1% P/S, 50 ng/ml mouse SCF (PeproTech, #250-03), 10 ng/ml mouse IL-3 (PeproTech, #213-13), and 10 ng/ml mouse IL-6 (PeproTech, #216-16). Cultures were performed in a 2 ml medium/well supplemented with or without a 10 mg/well Microniche at 37 °C with 5% $CO_2$ for 7 days.

For the long-term culture, $2 \times 10^5$ freshly isolated UCB CD34+ cells/well were cultured in a medium composed of StemSpan SFEMII, 1% P/S, 100 ng/ml hSCF, and 100 ng/ml hTPO. Cultures were performed in 2 ml medium/well supplemented with or without 40 nM UM171 or 10 mg/well Microniche at 37 °C with 5% $CO_2$. Cultured cells were collected and counted every 2 weeks. Meanwhile, complete medium changes were made and the cells continued to be cultured as described above.

For the cytokine rescue assay, $2 \times 10^5$ freshly isolated UCB CD34+ cells/well were cultured in a medium composed of StemSpan SFEMII, 1% P/S, 10 ng/ml hSCF, and 100 ng/ml hTPO. Cultures were performed in a 2 ml medium/well supplemented with or without a cytokine mixture [6.2416 ng/ml CXCL8 (SinoBiological, 10098-HNCH2), 0.2379 ng/ml CCL24 (SinoBiological, 10901-H08B), 0.0107 ng/ml IL-10 (SinoBiological, 10947-H07H), 0.0055 ng/ml IL-1β (SinoBiological, 10139-HNAE), and 0.0055 ng/ml IL-1α (SinoBiological, 10128-HNCH)] at 37 °C with 5% $CO_2$ for 7 days. The concentration of cytokines added was calculated based on the secretion gap between the medium and the Microniche in the results of the Luminex liquid suspension (Supplementary Table 6).

For the assessment of HSPCs inside and outside Microniche, cells outside Microniche were obtained by naturally standing the culture plate and aspirating the culture supernatant after 7 days of culture. Then, 2 ml PBS was added to the same culture well and cells inside Microniche were resuspended via gentle pipetting multiple times. The cells inside or outside Microniche were then filtered through a 30 μm sterile nylon membrane, respectively.

For the verification of the role of CXCR4 in expansion of HSCs by Microniche, $2 \times 10^5$ freshly isolated UCB CD34+ cells/well were cultured in a medium composed of StemSpan SFEMII, 1% P/S, 10 ng/ml hSCF, and 100 ng/ml hTPO. Cultures were performed in a 2 ml medium/well supplemented with or without 1 ng/ml or 10 ng/ml human CXCL12 protein (SinoBiological, 10118-HNAE), or in a 2 ml medium/well containing 10 mg/well Microniche with or without 0.04 nM or 0.08 nM LY2510924 [an antagonist that selectively blocks the binding of SDF-1 to CXCR4, $IC_{50} = 0.079$ nM (Selleck, S8505)] at 37 °C with 5% $CO_2$ for 7 days.

For all experiments, the cultured cells were collected by Sorvall ST 16 Centrifuge (Thermo Scientific) at $415 \times g$ for 10 min, and the live cells were counted using trypan blue and an automated cell counter (Bio-Rad, TC20). For the Microniche-based culture, the starting cell suspension in a 100 μl medium was added to the well and fused with Microniche for 2 h and later supplemented with a 1.9 ml medium before entering the cell incubator. During the harvest, cells cultured with Cytopore1, Cytodex3, Microcarrier W01 or Microniche were resuspended via gentle pipetting multiple times, and the cells were filtered with a 30 μm sterile nylon membrane.

## Scanning electron microscope (SEM) and co-immunofluorescent staining

HSPCs cultured in Microniche were visualized using SEM (FEI Quanta 200) and a multi-photon microscope (Nikon A1RMP). For SEM, cell-laden microcarriers were fixed in 2.5% glutaraldehyde (GA), and dehydrated with graded ethanol and then tertiary butanol. The dehydrated samples were dried to a critical point and sputter-coated with gold before SEM observation. For fluorescence imaging, cells on microcarriers were stained with Calcein AM (Kai Ji biology) according to manufacturers' instruction, before imaging with the fluorescence microscope. Fluorescence images were collected by NIS-Elements F Software (v4.00.06) and analyzed by ImageJ (v1.53c).

## Flow cytometry analysis and cell sorting

Cell phenotypes in fresh and cultured cells were measured using a combination of the fluorescent-labeled antibodies. Cells were stained in PBS supplemented with 2% FBS at 4 °C for 30 min, after which the stained cells were washed once with PBS. All the details of antibodies and reagents are listed in Supplementary Table 8 and were used in

1:100 or 1:1000 (DAPI solution) dilution according to manufacturer's recommendation. A FACS cantoII, LSRII, or FACS AriaIII (both from BD) instrument was used for flow cytometry analysis or cell sorting. Data were collected by BD FACSDiva Software (v7.0 and v8.0.1) and analyzed using FlowJo V10 Software (v10.6.2).

## Scalable culture

In total, $1.5 \times 10^8$ UCB MNC cells, freshly isolated from one single UCB unit each time, were cultured in each bottle for 3 days in a medium composed of StemSpan SFEMII, 1% P/S, 10 ng/ml hSCF, and 100 ng/ml hTPO in 125 ml 3D FloTrix®(CytoNiche Technologies) with 50 mg Microniche in each bottle. Bottles with 60 ml mixtures were cultured at 37 °C in a humidified incubator with 5% $CO_2$ in the air. After 3 days of 40 rpm dynamic bulk-culture, the total cells were collected and counted, and stained with antibodies CD34, CD38, CD45RA, CD90, CD49f, CD62L, and CD133 (Supplementary Table 8). Fluorescence signals were detected using an FACS AriaIII flow cytometer.

## Assessment of cell morphology

Cell morphology was assessed using slides prepared with a cytocentrifuge (Cytospin 4, Thermo Scientific) at 400 rpm for 5 min, followed by Wright–Giemsa staining (G1020, Solarbio Science & Technology). The brightfield slides were scanned with NanoZoomer S360 (Hamamastu Photonics) at 400x. The images were recorded and analyzed using NDP.view 2.9.22 RUO. Briefly, the diameter of 50 randomly selected cells from each group were averaged and then compared between the groups.

## CFU-Mk assay

For mouse CFU-Mk, $1.2 \times 10^6$ cultured c-Kit$^+$ cells were collected and cultured in 1.7 ml MegaCult-C medium (STEMCELL Technologies, #04900) supplemented with cytokines (50 ng/ml hTPO + 20 ng/ml hIL-6 + 10 ng/ml mIL-3) in IMDM and collagen solution to assess the colony formation of Mk progenitors ($3 \times 10^5$ cells for each chamber), according to the manufacturer's instructions. Colonies were scored after 6–8 days with an inverted microscope (Nikon Ti-U).

For human CFU-Mk, $2 \times 10^5$ cultured UCB CD34$^+$ cells were collected and cultured in a 2 ml MegaCult-C medium with cytokines (STEMCELL Technologies, #04901) supplemented with collagen solution to assess the colony formation of Mk progenitors ($5 \times 10^4$ cells for each chamber), according to the manufacturer's instructions. Colonies were scored after 10–12 days with an inverted microscope (Nikon Ti-U). Microscopically magnified colonies images were collected by NIS-Elements F Software (v4.00.06).

## Transplantation assays in immunodeficient mice

For limit dilution assay, freshly isolated UCB CD34$^+$ or expanded progeny were transplanted via tail vein injection in 200 μl PBS into irradiated (120 cGy administered at least 4 h before transplantation) NOG mice according to the dose gradient. Primary recipients were sacrificed after 16 weeks. Bone marrow was extracted from the femora, tibiae, and ilia, while half of the BM cells from each recipient were transplanted into an irradiated secondary recipient. The remaining BM cells were evaluated by flow cytometry to identify the percentage of human engraftment and multi-lineage reconstruction. Secondary engraftment was assessed after 16 more weeks. HSC frequencies were computed using extreme limiting dilution analysis (ELDA)[55].

For limit dilution of scalable culture, freshly isolated MNCs from one single UCB unit were divided equally into two parts, one of which was sorted out CD34$^+$ cells by magnetic CD34 MicroBeads as a Fresh group, and the other part was cultured in a large-scale stirred bioreactor with Microniche. CD34$^+$ cells from Fresh group or expanded MNCs progeny from Microniche group were transplanted via tail vein injection in 300 μl PBS into irradiated (120 cGy administered at least 4 h before transplantation) NCG mice according to the dose gradient.

Mice were sacrificed after 16 weeks and bone marrow of mice was extracted from the femora, tibiae, and ilia to identify the percentage of human engraftment and multi-lineage reconstruction by flow cytometry. HSC frequencies were computed using ELDA.

For human Mk-biased HSCs xenograft assay, fresh or cultured cells were collected and stained with a combination of antibodies. Indicated cells (CD49f$^{low}$, nonCD49f$^{low}$, CD62L$^-$CD133$^+$ and non-CD62L$^-$CD133$^+$) were sorted using a FACS AriaIII (BD). Mice were irradiated (120 cGy) at least 4 h in advance and were anesthetized with 150 μl 1.25% 2,2,2-Tribromoethanol (Sigma-Aldrich, T48402) via intraperitoneal injection before transplantation. The joints of the mouse femur and tibia were shaved to facilitate the intra-bone marrow injection. Cells in the marked well were transferred into a 29 g insulin syringe (BD, 328421) and injected into mouse tibiae. The mice were sacrificed after 16 weeks and BM was extracted for flow cytometry analysis, while PB was collected for CBC analysis.

## Complete blood counts (CBC)

Peripheral blood was collected from the tail vein of living mice and analyzed using an automated hematology analyzer (Sysmex, XN-9000) according to the manufacturer's instructions. The value of platelets was measured via the PLT-fluorescence (PLT-F) channel.

## BD Rhapsody Single-Cell Analysis

Approximately $2 \times 10^4$ CD34$^+$CD38$^-$CD45RA$^-$CD90$^+$ cells from each group were sorted using a FACS AriaIIII flow cytometer, after which we performed targeted single-cell RNA-Seq analysis using the BD Rhapsody Single-Cell Analysis System (BD Biosciences), according to the manufacturer's instructions. In short, the single-cell suspension was loaded into a BD Rhapsody cartridge (BD Biosciences, 0109076) and single-cell capture was achieved by random distribution and gravity precipitation. The cells were lysed in the microwell cartridge to hybridize mRNA molecules to barcoded capture oligos on the beads. The beads were then collected from the microwell cartridge into a single tube for subsequent DNA synthesis and multiplex-PCR-based library construction. For the library construction, we used the customized BD Rhapsody Immune Response Panel for humans (BD Biosciences), while oligonucleotide antibodies of CD34, CD38, CD45RA, CD90, and CD49f (Supplementary Table 8) were used in 1:25 dilution according to manufacturer's recommendation for the Ab-seq library. Sequencing on the Illumina Hiseq PE150 and matrix was completed by Novogene Co., LTD (Beijing, China). The results were analyzed and visualized using BD SeqGeq software (v1.6.0, BD Biosciences). This analysis identified 7907 cells with 28,465 features detected for the control sample and 8017 cells with 28,465 features detected for the Microniche sample.

## RNA sequencing

In total, $1 \times 10^6$ CD34$^+$ cells were prepared for sequencing. RNA extraction, library preparation, sequencing, and data analysis were completed by Novogene Co., LTD (Beijing, China). In total, 3 μg of RNA per sample from three replicates per group were used as the input material. Sequencing libraries were generated using an NEBNext® UltraTM RNA Library Prep Kit for Illumina® (NEB, USA) according to the manufacturer's instructions. Sequencing was performed on a cBot Cluster Generation System using a TruSeq PE Cluster Kit v3-cBot-HS (Illumia) according to the manufacturer's instructions. After mapping to the reference genome and quantification, DESeq2 (R package, v1.10.1) was used to analyze the differential expression genes. | Log$_2$(FoldChange)| >1 and $q$-value <0.05 were considered significant. The significantly changed genes were used in subsequent GO and KEGG enrichment analyses. GO term results of the molecule function (MF) were summarized and clustered based on semantic similarity measures using the online tool REVIGO[56]. The total gene list was used in GSEA analysis (GESA Software v4.1.0).

## RNA extraction and quantitative reverse transcriptase PCR

RNA was extracted using a RNeasy Micro Kit (QIAGEN), and its quantity and quality were assessed using a NanoDrop 3300 Fluorospectrometer (Thermo Fisher Scientific) or an Agilent 2100 Bioanalyzer (Agilent Technologies). For qPCR, reverse transcription of extracted RNA was performed using SuperScript II Reverse Transcriptase (Invitrogen), and qPCR was performed with PCR Power SYBR Green mix (Applied Biosystems) on a QuantStudio™ 6 Real-Time PCR Systems (Applied Biosystems) according to the manufacturer's instructions. Data were collected by QuantStudio™ Design & Analysis Software (v1.4.3). mRNA signals were normalized to the internal control GAPDH (glyceraldehyde-3-phosphate dehydrogenase). Primers for target genes are listed in Supplementary Table 9.

## Luminex liquid suspension chip detection

Luminex liquid suspension chip detection was performed by Wayen Biotechnologies (Shanghai, China). The Bio-Plex Pro Human Chemokine Panel 40-plex kit was used according to the manufacturer's instructions. The conditioned medium of each group with three repeats was incubated in 96-well plates embedded with microbeads for 1 h, and then incubated with detection antibodies for 30 min. Streptavidin-PE was then added into each well for 10 min, while the values were read using the Bio-Plex MAGPIX System (Bio-Rad).

## 10X Genomics single-cell RNA sequencing

Approximately $1 \times 10^4$ CD34$^+$ cells were prepared for sequencing. Library construction, sequencing, and matrix data were completed by Novogene Co., LTD (Beijing, China). For experiments using the 10X Genomics platform, the Chromium Single-Cell 3' Library & Gel Bead Kit v2 (PN- 120237), Chromium Single-Cell 3' Chip kit v2 (PN-120236), and Chromium i7 Multiplex Kit (PN-120262) were used according to the manufacturer's instructions in the Chromium Single-Cell 3' Reagents Kits v2 User Guide. Libraries were run on the Illumina Hiseq PE150 sequencing. Reads were aligned to GRCh38 reference assembly (v2.2.0, 10X Genomics). Post-processing and quality control were performed by Novogene using the 10X Cell Ranger package (v2.1.0, 10X Genomics). Subsequent analyses of cellranger reanalyze and Seurat were all performed based on this output gene expression matrix. The Seurat R-package (v4.0.1) was used to normalize data, dimensionality reduction, clustering, and differential expression. The data was dimension reduced using Uniform manifold approximation and projection (UMAP). Cluster annotation were based on their canonical marker genes, and was referred to CellMarker database[57]. This analysis identified 14,830 cells for the control sample and 16,744 cells for the Microniche sample. The mean number of reads per cell was 54,004 for the control sample and 42,983 for the Microniche sample. The median number of genes detected per cell was 2892 genes for the control sample and 2500 genes for the Microniche sample.

## Statistical analysis

All data are presented as a mean ± s.d. from $n \geq 3$ independent biological replicates unless otherwise stated. Graphing and statistical analyses were performed with software Graphpad Prism 9 and Microsoft Office Excel 2019. For limiting dilution analysis, statistical significance was determined with a chi-square test by ELDA software[55] (http://bioinf.wehi.edu.au/software/elda/). For in vitro culture of AA patient specimens, statistical significance was determined with a paired two-tailed Student's $t$ test. For the rest of the datasets, a two-tailed Student's $t$ test was used to assess the significance. The threshold for significance in all tests was $p < 0.05$. The specifics of the statistical tests and number of replicates are stated in the figure legends.

## Reporting summary

Further information on research design is available in the Nature Portfolio Reporting Summary linked to this article.

## Data availability

The raw sequence data reported in this paper have been deposited in the Genome Sequence Archive (Genomics, Proteomics & Bioinformatics 2021) in the National Genomics Data Center (Nucleic Acids Res 2022), China National Center for Bioinformation/Beijing Institute of Genomics, Chinese Academy of Sciences and are available under the accession number "HRA003310". Gene Set Enrichment analysis gene sets came from MSigDB collections of the GSEA gene set database with accession numbers "M9809", "M14297" and "M4406"). Canonical cell markers of interest in specific cell types (Supplementary Fig. 6f) came from the CellMarker database (http://bio-bigdata.hrbmu.edu.cn/CellMarker/). All other data supporting the results in this study are available within the paper and the Supplementary Information or from the corresponding authors upon reasonable request. Source data are provided with this paper.

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

## Acknowledgements

We thank Chao Wu (Zhejiang University), Qiqi Chu (BD Co. Ltd.), and Yibo Zhang (BD Co. Ltd.) for assistance with the analysis of single-cell sequence data. This work was supported by grants from the Ministry of Science and Technology of China (2017YFA0104900 to Y.G. and Y.D., and 2016YFA0100600 to Y.G.), the Haihe Laboratory of Cell Ecosystem Innovation Fund (HH22KYZX0002 to Y.G., and HH22KYZX0040 to Y.L.), the National Natural Science Foundation of China (92068204 to Y.G., Y.D., and J.S., 81870083 to Y.G., 81970105 to Y.L., and 82200126 to W.Z.), CAMS Innovation Fund for Medical Science (2021-I2M-1-019 to Y.G. and W.Z.) and a SKLEH-Pilot Research Grant.

## Author contributions

Y.G., Y.D., J.S., T.C. and Y.L. designed the research. Y.L., M.H. and Y.G. analyzed data. Y.L., M.H., Y.G., Y.D., J.S., W.Z. and W.L. wrote the paper. Y.L., M.H., Wen.L. and C.W. made the Figures. Y.D. and W.L. designed and produced Microniche. W.Z. constructed library of single-cell sequencing and the large-scale culture. M.Y. and M.H.

injected cells in animal experiments. Z.W. performed CellphoneDB analysis. M.H., W.Z., H.X., Y.L., H.Z., H.L., W.L., W.F., S.X., X.L., S.F., L.Z., C.W., Le.Z., Ya.L., J.G., J.Y., Y.Z., Y.X. and X.M. performed other experiments.

## Competing interests

The authors declare no competing interests.
