## [Peer Review File · Nature Communications]

REVIEWER COMMENTS

Reviewer #2 (Remarks to the Author):

In this paper, Li et al. presented a strategy for human HSC expansion based on a biomimetic microcarrier “Microniche” (Beijing CytoNiche Biotechnology Co., Ltd.). By using in vitro cultures, xenograft models, and flow-cytometry immunophenotyping, the authors demonstrated the expansion of HSCs, particularly of a megakaryocytes-biased HSC subpopulation, derived from different sources (UCB, PB, BM). The topic of the work is interesting as the expansion of functional HSCs is an essential point to exploit the full potential of HSC-based therapies. The authors presented a big amount of data, which is mostly descriptive. This makes it difficult to fully understand the scope of the work (expansion of functional HSCs for transplant purposes, characterization of Mk-biased HSC populations, mechanistic studies on HSC expansion?).

Specific points to address are:

To evaluate the effects of the Microniche culture, the authors used as a control condition an in vitro culture with “conditioned medium only”. However, a clear description of what the authors mean by the conditioned medium is needed. Is the conditioned medium deriving from the Microniche cultures? If yes, is the medium conditioned in the absence or presence of cells? In the latter case, it might reduce the significance of the effect of the cytokine milieu emphasized by the authors.

In figure 1C-D, the gating strategy used should be better clarified. For example, in the text, the authors mention the CD34+CD38-CD45RA-CD90+CD49f+ cell population, but in the Figures, it does not appear clear whether the gated cells belong all to the same population. If this is the case, the dot plot should be connected with arrows (as it was done in Supplemental Figure 3D).

In Figure 1H authors transplanted Fresh and Microniche expanded bone marrow MNCs derived from patients affected by Aplastic Anemia. After 16 weeks 0.061% of bone marrow cells are hCD45+/mCD45-. Is this percentage sufficient to demonstrate the engraftment? Parallel experiments should be performed with healthy donor-derived bone marrow MNC. In Supplemental Figure 2C, the same experiment has been performed with UCB-derived CD34+ cells and the percentages of hCD45+/mCD45- were around 2000 times higher. Please comment.

It is not clear the HSC cell source used for the experiments performed in Supplemental Figures

3C, 3D and Figures 2A, B, C, D, E, F. Throughout the entire manuscript the succession of the experiments should be better organized (i.e. taking into account the cell source) to allow a clear comparison among results.

To assess the successful engraftment of transplanted HSCs and the multilineage reconstruction, the authors should also analyze peripheral blood cell populations and not only bone marrow cell populations.

Regarding the assessment of successful engraftment and reconstruction, the authors used a threshold of 0.01% of cells (i.e. Figure 4F). Is not this value too low to assess successful engraftment? Please comment.

Regarding the effect of properly mimicking the bone marrow niche as a pivotal contributor to the expansion of Mk-biased HSCs obtained with Microniche cultures, the data presented are quite scarce to sustain it. The authors compared Microniche with other biomimetic microcarriers arguing that the physical structure of Microniche was essential for the expansion of primitive HSCs, especially for Mk-biased HSCs. To clearly demonstrate it, the authors should perform experiments in which the two physical parameters are taken into consideration (porosity and elasticity) are singularly modified.

The authors showed that cells retained inside the Microniche have high CXCR4 expression. These data are only descriptive. Does it have a role in HSC expansion by Microniche? The authors might perform experiments with CXCR4 inhibitors or try to displace CXCR4 highly expressing cells from the inside of Microniche by using gradients of SDF-1 in the culture medium.

Reviewer #2 (Remarks to the Author):

In this paper, Li et al. presented a strategy for human HSC expansion based on a biomimetic microcarrier "Microniche" (Beijing CytoNiche Biotechnology Co., Ltd.). By using in vitro cultures, xenograft models, and flow-cytometry immunophenotyping, the authors demonstrated the expansion of HSCs, particularly of a megakaryocytes-biased HSC subpopulation, derived from different sources (UCB, PB, BM). The topic of the work is interesting as the expansion of functional HSCs is an essential point to exploit the full potential of HSC-based therapies. The authors presented a big amount of data, which is mostly descriptive. This makes it difficult to fully understand the scope of the work (expansion of functional HSCs for transplant purposes, characterization of Mk-biased HSC populations, mechanistic studies on HSC expansion?).

Specific points to address are:

1. To evaluate the effects of the Microniche culture, the authors used as a control condition an in vitro culture with "conditioned medium only". However, a clear description of what the authors mean by the conditioned medium is needed. Is the conditioned medium deriving from the Microniche cultures? If yes, is the medium conditioned in the absence or presence of cells? In the latter case, it might reduce the significance of the effect of the cytokine milieu emphasized by the authors.

Response: In this study, we employed SFEMII + 1% P/S + 10 ng/mL hSCF + 100 ng/mL hTPO as a conditioned medium (detailed further in the methods section, lines 438-488). Two cytokines, hSCF and hTPO were added to the serum-free medium, without any medium derived from Microniche cultures or scaffolding cells. In response to your comment, we have added additional clarification to our text (line 88) and figure legend (line 810-811).

2. In figure 1C-D, the gating strategy used should be better clarified. For example, in the text, the authors mention the CD34+CD38-CD45RA-CD90+CD49f+ cell population, but in the Figures, it does not appear clear whether the gated cells belong all to the same population. If this is the case, the dot plot should be connected with arrows (as it was done in Supplemental Figure 3D).

Response: Thank you for your suggestion. In response, we supplemented the gating strategy for the *in vitro* culture in Supplemental Figure 1e.

Figure S1e:

3. In Figure 1H authors transplanted Fresh and Microniche expanded bone marrow MNCs derived from patients affected by Aplastic Anemia. After 16 weeks 0.061% of bone marrow cells are hCD45+/mCD45-. Is this percentage sufficient to demonstrate the engraftment? Parallel experiments should be performed with healthy donor-derived bone marrow MNC. In Supplemental Figure 2C, the same experiment has been performed with UCB-derived CD34+ cells and the percentages of hCD45+/mCD45- were around 2000 times higher. Please comment.

Response: As we known, a profound defect within the stem cell compartment is a unifying feature across most AA patients (Maciejewskia et, al. Archives of Medical Research, 2003; Young, The new England journal of medicine, 2018). As a stem cell disease, AA may reflect the function and quantity of normal hematopoietic stem cells and their ability to regenerate. The differentiation of AA patient cells could not be observed in xenografts, which demonstrated that the stemness of AA stem cells was not completely restored after an *in vitro* treatment, but was still stronger than original cells and were capable of being implanted.

4. *It is not clear the HSC cell source used for the experiments performed in Supplemental Figures 3C, 3D and Figures 2A, B, C, D, E, F. Throughout the entire manuscript the succession of the experiments should be better organized (i.e. taking into account the cell source) to allow a clear comparison among results.*

Response: Thank you for your suggestion. In response, we have included notes on these figures and legends to clarify the information (in Figure S3C, S3D and Figure 2A, B, C, D, E, F and the corresponding legends). We have highlighted these words in the revised manuscript.

5. To assess the successful engraftment of transplanted HSCs and the multilineage reconstruction, the authors should also analyze peripheral blood cell populations and not only bone marrow cell populations.

Response: Thank you for your suggestion. In this study, we conducted three xenograft assays (LDA of cultured cells in plate, LDA of scalable expansion, and LDA of sorted Mk-biased HSCs). Thereafter, we analyzed the reconstruction of human cells in PB. We noted reconstruction of human CD45+, CD45+CD33+, CD45+CD19+, and CD41+ cells in assays from culture in the plates (shown in Figure S2c) and bioreactor (shown in Figure S4i), but reconstruction was absent in the LDA of sorted Mk-biased HSCs (data not shown). Because the number of total cells harvested from PB was much lower than that harvested from BM, we used the flow cytometry data from the BM cells in the HSC frequency calculating.

Figure S2c:

Figure S4i:

6. Regarding the assessment of successful engraftment and reconstruction, the authors used a threshold of 0.01% of cells (i.e. Figure 4F). Is not this value too low to assess successful engraftment? Please comment.

Response: In Figure 4, we used sorted Mk-biased HSCs for xenograft assays. From our experience, the reconstruction levels in mice receiving human cells were much lower when using only sorted HSCs (such as CD34+CD38-CD45RA-CD90+ cells) than when using CD34+ HSPCs for transplantation. As mentioned previously, we found only a few recipients reconstructed human hematopoiesis in PB. However, we noted human CD45+ and CD41+ cell populations in BM. These populations were continuous with the negative population, but had obvious positive fluorescence intensities. Thus, we used 0.01% as the threshold in assessment of successful reconstruction. Using this standard, our data still had statistical differences between CD62L-CD133+ and non- CD62L-CD133+ groups. For this reason, we considered that the reconstructions in mice receiving human cells were much lower when using only sorted HSCs. Moreover, more suitable strains of immune-deficient mice for reconstruction by sorted HSC should be developed and used in the future. We also discussed these limitations in the revised manuscript (line 366-370).

7. Regarding the effect of properly mimicking the bone marrow niche as a pivotal contributor to the expansion of Mk-biased HSCs obtained with Microniche cultures, the data presented are quite scarce to sustain it. The authors compared Microniche with other biomimetic microcarriers arguing that the physical structure of Microniche was essential for the expansion of primitive HSCs, especially for Mk-biased HSCs. To clearly demonstrate it, the authors should perform experiments in which the two physical parameters are taken into consideration (porosity and elasticity) are singularly modified.

Response: Thank you this insight and suggestion. Indeed, we measured the pore size of ECM network in published immunofluorescence staining images of bone marrow matrix (Coutu DL., et al., Nat Biotechnol, 2017; Gomariz A., et al., Nat Commun, 2018) and determined that the pore size of bone marrow ECM varies based on to internal position. The matrix pore size of metaphysis, cortical bone, and the central area of bone marrow were 30-100 μm , 3-19 μm , and 5-30 μm , respectively. Therefore, a pore size of 20-30 μm for microcarriers (Cytopore 1, Microcarrier W01, and Microniche) was chosen to represent the pore size of the bone marrow matrix in the center. We determined that a change to the pore size was not necessary because pore size of the BM has a wide range.

After the aperture parameter range was fixed, we explored the regulation of microcarrier elasticity on HSCs. Accordingly, based on young's bone marrow modulus data in published studies (Chen X., et al., Biophys J, 2020; Vining KH, Mooney DJ., et al., Nat Rev Mol Cell Biol, 2017), the elasticity of the matrix in the center of the bone marrow was 0.3-24.7 kPa. Therefore, the 20 kPa (within the range, similar to the bone marrow, Microniche) and 40 kPa (outside the range, Cytopore 1 and Microcarrier W01) levels of elasticity of microcarriers were selected to explore the regulation of elasticity of microcarriers on HSCs. We found that properly simulating the BM niche's physical structure (porosity and elasticity) by the Microniche was essential for expansion of primitive HSCs.

More importantly, constructing microcarriers with large differences compared to the young's modulus or pore size necessitates a change to the synthetic processing of microcarriers. This change will bring new parameter changes leading to the failure of designs. Using our current data demonstrating that only Microniche was more appropriate for the parameters of BM, we also obtain the conclusion that proper biomimicry contributes to the expansion of human HSCs.

8. The authors showed that cells retained inside the Microniche have high CXCR4 expression. These data are only descriptive. Does it have a role in HSC expansion by Microniche? The authors might perform experiments with CXCR4 inhibitors or try to displace CXCR4 highly expressing cells from the inside of Microniche by using gradients of SDF-1 in the culture medium.

Response: Thank you for your suggestion. Here, we employed CXCR4 as the marker of bone marrow-retained cells. Our data supported the conclusion that the cells inside of Microniche were more like cells inside BM.

Based upon your suggestion, we added supplemental experiments on CXCL12 to control medium and supplementation of CXCR4 inhibitor LY2510924 in the Microniche group to assess the role of CXCR4 in expansion of HSCs. Our data demonstrated that addition of CXCL12 was not able to increase primitive HSCs compared to culture using only control medium. The blocking of CXCR4 significantly decreases the expansion effect of primitive HSCs by Microniche. These data suggested that physical scaffolding was essential for the expansion of Mk-biased HSCs.

Figure S8

REVIEWERS' COMMENTS

Reviewer #1 (Remarks to the Author):

The authors have answered most of the reviewers' questions.